# Seasonal streamflow forecasts for Europe – I. Hindcast verification with pseudo- and real observations

Wouter Greuell[1], Wietse H. P. Franssen[1], Hester Biemans[2] and Ronald W. A. Hutjes[1,2]

1) Water Systems and Global Change, Wageningen University, Droevendaalsesteeg 3, NL 6708 PB Wageningen, Netherlands
2) Water and Food, Wageningen Environmental Research, Droevendaalsesteeg 3, NL 6708 PB Wageningen, Netherlands

correspondence to wouter.greuell@wur.nl

**Abstract**

Seasonal predictions of river flow can be exploited among others to optimize hydropower energy generation, navigability of rivers and irrigation management to decrease crop yield losses. This paper is the first of two papers dealing with a physical model-based system built to produce probabilistic seasonal hydrological forecasts, applied here to Europe. This paper presents the development of the system and the evaluation of its skill. The Variable Infiltration Capacity (VIC) hydrological model is forced with bias-corrected output of ECMWF's Seasonal Forecasting System 4. For the assessment of skill, we analysed hindcasts (1981-2010) against a reference run, in which VIC was forced by gridded meteorological observations. The reference run was also used to generate initial hydrological conditions for the hindcasts.

The skill in runoff and discharge hindcasts is analysed with monthly temporal resolution, up to 7 months of lead time, for the entire annual cycle. Using the reference run output as pseudo-observations and taking the correlation coefficient as metric, hot spots of significant theoretical skill in discharge and runoff were identified in Fennoscandia (from January to October), the southern part of the Mediterranean (from June to August), Poland, northern Germany, Romania and Bulgaria (mainly from November to January), western France (from December to May) and the eastern side of Great Britain (January to April). Generally, the skill decreases with increasing lead time, except in spring in regions with snow-rich winters. In some areas some skill persists even at the longest lead times (7 months).

Theoretical skill was compared to actual skill as determined with real discharge observations from 747 stations. Actual skill is generally substantially less than theoretical skill. This effect is stronger for small than for large basins. Qualitatively, the use of different skill metrics (correlation coefficient, ROC area and Ranked Probability Skill Score) leads to broadly similar spatio-temporal patterns of skill, but the level of skill decreases, and the area of skill shrinks, in the following order: correlation coefficient, ROC area below normal tercile, ROC area above normal tercile, Ranked Probability Skill Score and finally, ROC near normal tercile.

# 1    Introduction

Society may benefit from seasonal hydrological forecasts, i.e. hydrological forecasts for future time periods from more than two weeks up to about a year (Doblas-Reyes et al., 2013). Such predictions can e.g. be exploited to optimize hydropower energy generation (Hamlet et al. 2002), navigability of rivers in low flow conditions (Li, et al., 2008) and irrigation management (Ghile and Schulze 2008; Mushtaq et al. 2012) to decrease crop yield losses. In order to be of any value in decision making processes in such sectors, forecasts must be credible, i.e. be skilful in predicting anomalous system states, as well as being relevant and legitimate to the decision making process (e.g. Bruno Soares and Dessai, 2016). In this paper we will introduce WUSHP (Wageningen University Seamless Hydrological Prediction system), a dynamical, model-based system (see Yuan et al., 2015) that was built around the Variable Infiltration Capacity (VIC) hydrological model and ECMWF's Seasonal Forecast System 4, to produce seasonal hydrological forecasts. It will be applied to Europe. The usefulness of the system depends partially on the level of its skill and the paper will therefore focus on an extensive assessment of the skill of WUSHP. The usual method of assessing skill of predictive systems is by analysing hindcasts, a strategy that will be adopted here as well.

During recent years, a number of systems for making seasonal hydrological forecasts have been developed. Examples are the University of Washington's Surface Water Monitor (SWM; Wood and Lettenmaier, 2006) and the African Drought Monitor (Sheffield et al., 2014). Seasonal hydrological forecasting systems for the entire continent of Europe are scarce (Bierkens and van Beek, 2009; Thober et al., 2015), but a few more concentrate on smaller domains such as the British Isles (Svensson et al., 2015), Iberia (Trigo, 2004) or France (Céron et al., 2010; Singla et al., 2012).

Thober et al. (2015) forced a mesoscale hydrological model (mHM) with meteorological hindcasts from the North American Multi-Model Ensemble (NMME) to investigate the predictability of soil moisture in continental Europe, excluding Fennoscandia. Evaluating at seasonal resolution a number of forecasting techniques that produced distinct variations in the magnitude of skill, they found that spatial patterns in skill were remarkably similar among the different techniques, as well as comparable to the spatial patterns of the autocorrelation (persistence) of reference soil moisture. High skill was found in eastern Germany and Poland, Romania, the southern Balkans and eastern Ukraine as well as north-western France. Less skill was found in the mountainous areas of the Alps and the Pyrenees, the northern Adriatic and Atlantic Iberia. Most skill was found for winter months (DJF), least for autumn (SON), this minimum shifting to summer (JJA) at long lead times (6 months).

Bierkens and van Beek (2009) developed an analogue events method to select annual ERA40 meteorological forcing on the basis of annual SST anomalies in the northern Atlantic and then made hydrological forecasts with a global-scale hydrological model applied to Europe. Evaluating only winter and summer half year aggregated skill for

discharge, they found wintertime skill in large parts of Europe with maxima in eastern Spain and a zone from the southern Balkans and Romania through eastern Poland and western Russia to the Baltic states and Finland. Summertime skill was lower, generally by about 50% and even more around the Alps and the Adriatic. A climate forecast based on the North Atlantic Oscillation (NAO) added significant skill only in limited areas, such as Scandinavia, the Iberian Peninsula, the Balkans, and around the Black Sea.

Svensson et al. (2015) found skilful winter river flow forecasts across the whole of the UK due to a combination of skilful winter rainfall forecasts for the north and west, and strong persistence of initial hydrological conditions in the south and east. Strong statistical correlations between the NAO index and winter precipitation in Iberia lead to skilful forecasts of JFM river flow and hydropower production (Trigo et al., 2004). Céron et al. (2010) and Singla et al. (2012) set up a high resolution river flow forecasting system (8 km) over France, for which the seasonal climate forecast improved the MAM skill over northern France, but worsened it over southern France (compared to a river flow model with proper initialisation of soil moisture, snow etc., but random atmospheric forcing). Demirel et al. (2015) found that both two physical models and one neural network over-predict runoff during low-flow periods using ensemble seasonal meteorological forcing for the Moselle basin. As a result forecasts of more extreme low flows are less reliable than forecasts of more moderate ones.

It is quite common in seasonal hydrological forecasting (e.g. Shukla and Lettenmaier, 2011, Singla et al., 2012, Mo and Lettenmaier, 2014, and Thober et al., 2015) but also in medium range forecasting (i.e. 14 days in Alfieri et al., 2014) to determine prediction skill by comparing the hindcasts with the output from a reference simulation. A reference simulation is a simulation made with the same hydrological model as the hindcasts, except that the forcing is taken from meteorological observations or from a gridded version of meteorological observations. The reference simulation can best be regarded as a simulation that attempts to make a best estimate of the true conditions (in terms of e.g. discharge, soil moisture and evapotranspiration), using the modelling system. We will refer to the output of such a reference simulation as "pseudo-observations" (alternatively named "true discharge" in Bierkens and Van Beek, 2009; "synthetic truth" in Shukla and Lettenmaier, 2011; "reanalysis" in Singla et al., 2012; "a posteriori estimates" in Shukla et al., 2014). We prefer the term "pseudo-observations" over "re-analysis" since the latter has a meteorological connotation that often implies the use of some form of (variational) data assimilation. We did not attempt any form of assimilating observed hydrological variables, such as discharge, in our reference run.

Pseudo-observations have the important advantages of being complete in the spatial and the temporal domain and to be available for all model variables. Also, they are suitable for the quantification of small sensitivities, e.g. to bias correction of the meteorological forcing, which would be hard to detect with real observations. Finally, assessment of skill based on pseudo-observations reduces model errors from the analysis to a

minimum, which is especially useful when addressing various sources of skill (Wood et al., 2016), something we will do in the companion paper (Greuell et al., 2016, in revision).

The downside of pseudo-observations is, of course, that they are not equal to real observations. In this paper we will determine the performance of the prediction system not only with pseudo-observations, but also with real observations of discharge (like e.g. Koster et al., 2010, and Yuan et al., 2013) and compare the skill found with the two different approaches ("theoretical and actual skill", according to Van Dijk et al., 2013). Such a comparison was previously made by Bierkens and Van Beek (2009) and Van Dijk et al. (2013) and they found that theoretical skill generally exceeds actual skill. This is in line with the fact that the pseudo-observations are obtained with the same model as the hindcasts, which should logically lead to an overestimation of the skill when the pseudo-observations are used for verification. We thus hypothesise that theoretical skill exceeds actual skill. In this paper we will not only analyse the difference between the skills obtained with the two different types of data but also discuss in some detail conceptual differences between using pseudo- and real observations for verification.

This paper aims to analyse to what extent WUSHP is able to predict runoff and discharge in Europe over the full annual cycle and for lead times up to 7 months. We aim to assess skill at monthly resolution instead of seasonal or semi-annual aggregates. Where many studies use correlation coefficient as main skill metric we will also assess skill using two probabilistic metrics, namely ROC area and RPSS (see Sect. 2.3). The second aim of the paper is to get a better understanding of the effects of using pseudo-observations, as opposed to using actual observations, for the verification of hindcasts. In the next section we describe the concept and details of our modelling (Sect. 2.1) and analysis approach (Sect. 2.2 and 2.3). We will start the result section by assessing theoretical skill of the runoff hindcasts (Sect. 3.1) and then proceed to theoretical skill of the discharge hindcasts and a comparison between theoretical skill of discharge and runoff in Sect. 3.2. Differences between theoretical and actual skill of discharge will be presented (Sect. 3.3) followed by an analysis of differences in skill determined with various metrics in Sect. 3.4. The discussion starts with a conceptual analysis of reasons for differences in actual and theoretical skill (Sect. 4.1), followed by a discussion of uncertainties (Sect. 4.2) and implications (Sect. 4.3).

In a companion paper (Greuell et al., 2016) we analyse the reasons for the presence or lack of skill discussed in the present paper, using two different methods. Firstly, skill in the forcing and other directly related hydrological variables, like evapotranspiration, are analysed. Secondly, a number of experiments similar to the conventional Ensemble Streamflow Prediction (ESP) and reverse-ESP experiments, which isolate different causes of predictability, are discussed. In the results and discussion sections of the present paper we will occasionally look forward to the identified causes of skill.

180

181

## 2    System, models, data and methods of analysis

183

In the following subsections we will describe the various components of WUSHP (2.1), the real discharge observations (2.2) and the methods of analysis (2.3). Fig. 1 provides an outline of the system, which consists of the hindcasts themselves (middle box in the figure) and a model reference run (lower box). The hindcasts will be verified by means of the pseudo-observations, which are generated by the reference simulations, and by real discharge observations, which are "generated" in the real world (upper box). Differences between these two types of verifications will be discussed in Sect. 4.1.

191

192

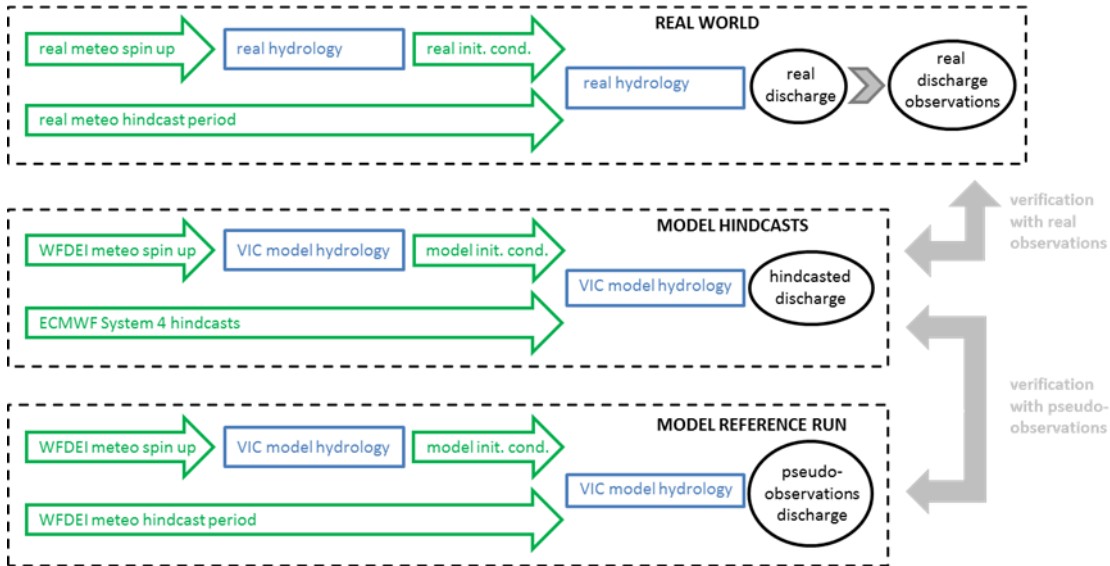

193

194

Figure 1:    Setup of the present study. The lower two dashed boxes summarise the setup of the forecast system itself. The upper dashed box represents the real world. The filled arrows on the right hand side represent verification of hindcasts (in the middle) with pseudo-observations (bottom) and with observations of real discharge (top). In each box the flow at the upper left represents the creation of initial conditions while the flow below that (a single arrow) represents the meteorological forcing.

202

203

## 2.1    The model, workflow and forcing data for the hindcasts and the reference simulation

206

WUSHP consists of two simulation branches: a single reference simulation (lower box in Fig. 1) and the hindcasts themselves (middle box in same figure). In both branches, terrestrial hydrology is simulated with the Variable Infiltration Capacity model (VIC, see Liang et al., 1994), which runs on a domain extending from 25˚ W to 40˚ E and from

35° to 72° N, including 5200 land based cells of 0.5° x 0.5° (see maps in e.g. Fig. 2). VIC is forced by a gridded data set of daily meteorological data (7 variables: precipitation, minimum and maximum temperature, atmospheric humidity, wind speed and incoming short- and long wave radiation).

In the reference simulation VIC is forced by the WATCH Forcing Data Era-Interim (WFDEI; Weedon et al., 2014) for the period of 1979-2010, of which the first two years were used to spin up the states of snow, soil moisture and discharge, and were not used in further analysis. The reference simulation has the dual aim to create the pseudo-observations for verification purposes (lower box in Fig. 1) and to create a best estimate of the temporally varying model state, which is then used for the initialisation of the hindcasts (flow from the upper left in de middle box of Fig. 1).

The second branch, the hindcasts, consists of three steps. Seasonal predictions of the same set of 7 meteorological variables (see above) are taken from ECMWF's Seasonal Forecast System 4 (S4 hereafter) at daily resolution. These are then bias-corrected using WFDEI as the reference data set. Finally, VIC is run with the bias-corrected S4 hindcasts as forcing, taking initial states from the reference simulation.

The S4 hindcasts used in the present study include 15 members, cover the period from 1981 to 2010 and consist of simulations with a duration of 7 months, starting and initialised on the first day of every month (see Molteni et al., 2011 and the ECMWF Seasonal Forecast User Guide, online). The S4 ensemble is constructed by combining a 5-member ensemble analysis of the ocean initial state with SST perturbations of that state and with activation of stochastic physics.

All seven meteorological forcing variables were regridded with bi-linear interpolation from the 0.75 x 0.75° lat-lon grid of the S4 hindcasts to a 0.5° x 0.5° grid. Since bias correction generally improves forecasting skill, the quantile mapping method of Themeßl et al. (2011) was applied to bias-correct the forcing variables, taking the WFDEI as reference. For each variable and grid cell, 84 correction functions were established and applied by separating the data according to target month (12) and lead month (7). Such empirical distribution mapping of daily values has been successful in improving especially forecast reliability (rather than sharpness and accuracy; Crochemore et al., 2016).

VIC was run for the period of the S4 hindcasts (1981 – 2010). Additionally, for the reference simulation two extra years (1979 – 1980) were simulated to spin up the states of snow, soil moisture and discharge. The hindcast simulations were initialised with states of soil moisture and snow from the reference simulation, so for these variables spin up was not needed. However, due to the set-up of the routing module of VIC, the state of discharge could not be saved and loaded. Hence to spin up discharge, each 7-month hindcast simulation was preceded by one month simulation with WFDEI forcing.

Since the hindcasts cover 30 years with 12 initialisation dates each and consist of 15 members, a total of 5400 hindcast simulations was carried out.

VIC is run in so-called 'energy balance mode' which requires resolving the diurnal cycle. Therefore, internally the model temporally disaggregates the daily input to 3-hourly data and runs with a time step of 3 hours. The output of all variables is again at a daily resolution. Because snow may contribute significantly to the seasonal predictability of other hydrological variables, VIC was run with the option of subgrid elevation bands. This means that for each grid cell calculations were carried out at up to 16 different elevations, with the aim of simulating the elevation gradient of snow. VIC was run in naturalised flow mode, i.e. river regulation, irrigation and other anthropogenic influences were not considered.

Simulations of historic discharge made with VIC (and four other hydrological models) were validated with observations from large European rivers by Greuell et al. (2015) and Roudier et al. (2016). VIC exhibits a fairly small average bias (across 46 stations) of +23 mm/yr (= 7%) and overall differentiates well between low and high specific discharge basins with a spatial correlation coefficient of 0.955. However, specific discharge is overestimated in the Mediterranean and underestimated in northern Fennoscandia. Annual cycles are fairly well reproduced across Europe, though VIC somewhat overestimates their amplitude. In northern Fennoscandia the spring peak is too late and lasts too long. Annual cycles are best reproduced for rain-fed rivers in central Europe while those for rivers with significant snow dynamics are good (Alps). However, the annual cycle is more poorly reproduced in basins with strong soil freezing dynamics (northern Fennoscandia) or strong damping of discharge amplitudes by large lakes (southern Finland).

Perhaps more relevant in the present context is the model's capability to reproduce inter-annual variations in discharge. On average across 22 discharge stations, the standard deviation of simulated annual discharge was 9% higher than observed and the spatial correlation coefficient between the two 0.94. Like most models, VIC is better in simulating high flows (95 percentile: Q95) than low flows (Q5); the first is slightly overestimated, the second more seriously underestimated. The inter-annual variation in Q5 is overestimated in central Europe and the Alps, but underestimated in Fennoscandia (overall spatial correlation coefficient across Europe 0.40). The inter-annual variation in Q95 shows no clear spatial pattern and the overall spatial correlation coefficient is 0.70.

All validation results discussed in these two paragraphs are for the VIC model forced by E-OBS (v9, Haylock et al. 2008). Our forcing, WFDEI, shows higher precipitation (+104 mm/yr) across most of Europe, except for the Alps, Scotland and westernmost Norway. According to Greuell et al. (2015) this leads to higher mean discharge, higher inter annual variability and higher Q95 (but not Q5) of simulated discharge for almost all stations.

## 2.2 Discharge observations

For the assessment of skill with real discharge observations, two data sets with daily resolution were acquired from the Global Runoff Data Centre, 56068 Koblenz, Germany (GRDC): the GRDC data set and the European Water Archive (EWA) data set. We mapped these two station data sets onto the VIC grid with its resolution of 0.5˚ x 0.5˚ and aggregated the daily data at a time step of a month. To enable the investigation of the effect of basin size on some of our results, we made two sub-classes of observations. The first comprised observations for basins larger than 9900 km$^2$ ("large basins"), the second contained basins smaller than the area of the grid cells, i.e. smaller than about 2530 km$^2$ in southern Europe (at 35º N) and 1050 km$^2$ at 70º N ("small basins").

Initially, in many cases the location of observation stations did not match with the corresponding river in the digital river network used in the routing calculations (DDM30, see Döll and Lehner, 2002). We corrected for this issue by matching the observations with the simulations by means of basin size. The size of the model basins ("model basin area") was determined by the DDM30 network. The size of the basins upstream of the observation stations ("station basin area") was taken from the meta data of the observations. First the station basin area was compared to the model basin area of the cell that is nearest to the station ("nearest model cell basin area"). After this first step the mapping procedure for each observation differed between the two classes of basins.

For large basins we proceeded as follows:
- If the station and the nearest model cell basin area differed by less than 15%, the observations were matched with the model calculations for the nearest model cell.
- Otherwise, the station basin area was compared with the model basin area of the eight cells surrounding the nearest model cell.
- The minimum of the eight differences was determined.
- If that minimum was less than 15%, the simulations for the corresponding cell were matched with the observations.
- Otherwise, the station was discarded.

For small basins we proceeded as follows:
- If the nearest model cell did not have an influx from any of the neighbouring cells, its simulations were matched with the observations.
- Otherwise, all of the eight neighbouring cells without influx were selected.
- Their simulations were averaged and matched with the observations.

We further discarded all observations with less than 21 years of data within the simulation period (1981-2010) for any of the months of the year. The final data set within our European domain contained 111 cells with observations for large basins and 636 cells with observations for basins smaller than a model grid cell.

These data sets do not include any variable or parameter characterising the level of
human impact. To enable analysis of the effect of anthropogenic flow modifications on
predictive skill, we quantified the human impact by performing two model simulations
with the Lund-Potsdam-Jena managed Land (LPJmL) model (Rost et al., 2008;
Schaphoff et al., 2013). This model was operated at the same spatial resolution (0.5° x
0.5°) and with the same river network (DDM30) as VIC, but LPJmL does include dams
(GRanD database; Lehner et al;. 2011) and associated reservoir management. From the
discharge output of a naturalized LPJmL run and an LPJmL run with reservoir operation
and irrigation, the human impact at cell level was quantified by computing the so-called
Amended Annual Proportional Flow Deviator (AAPFD; see Marchant and Hehir, 2002).
For the analysis in Sect. 3.3, we selected all discharge observations for large basins with
an AAPFD < 0.3, i.e. basins with a relatively small degree of human impact (about half
of all 111 basins).

## 2.3   Methods of analysis

From the model output, consisting of daily means, monthly mean values were computed,
which were then used for the analysis. The analysis is restricted to runoff, defined here
as the amount of water leaving the model soil either along the surface or at the bottom,
and discharge, defined here as the flow of water through the largest river in each grid
cell. Discharge accumulates all runoff from cells that are upstream in the model river
network, with delays due to transport inside cells and through the river network. Hence,
whereas runoff represents only local hydrological processes, discharge aggregates
hydrological processes occurring in the entire basin upstream of a particular cell.
Instead of analysing skill per target season and/or for a number of consecutive lead
months, we analysed skill for every combination of the 12 target and the 7 lead months.
The thus achieved higher temporal resolution of the skill metrics enables a more
accurate determination of the beginning and end of periods of skill. Moreover, skill at a
monthly resolution provides the possibility to determine the consistency of the skill
where we define consistent skill as skill that persists during at least two consecutive
target or lead months. In accordance with Hagedorn et al. (2005) we designated the first
month of the hindcasts as lead month zero, so target month number is equal to the
number of the month of initialisation plus the lead month number.
Three skill metrics (see Mason and Stephensen, 2008, for a good discussion of the why
and how of these) were computed for each target and lead month separately: i) the
correlation coefficient between the observations and the *median* values of the hindcasts
(referred to as "correlation coefficient" or R), ii) the area beneath the Relative Operating
Characteristics (ROC) curve (shortly "ROC area") and iii) the Ranked Probability Skill
Score (RPSS). The ROC area is computed for three categories of the observations and
hindcasts with an equal number of values, namely the categories containing the one third

highest, lowest and the remaining values (upper, lower and middle tercile, resp.; above, below and near-normal, AN, BN and NN categories). The same subdivision of observations and hindcasts in terciles was made to compute the RPSS. Since none of these metrics is sensitive to systematic biases in the forecasting system, no attempt was made to correct simulated runoff or discharge for any such errors prior to computing the skill metrics. So we focus our evaluation on the models capability to predict river flow anomalies rather than absolute river flows.

All three skill metrics quantify, though in different ways, how well the ranking of the hindcasts matches the ranking of the observations. The correlation coefficient is a measure of the association between (pseudo-) observation and forecast ensemble median; we used the Pearson correlation coefficient. The ROC area is a measure of resolution or discrimination and indicates whether the forecast probability of an event (i.e. value falling in the considered tercile) is higher when such an event occurs compared to when not. The RPSS is a measure of accuracy and summarizes in a single number the skill of a forecast system to make forecasts with the correct percentage of ensemble members falling in any of the defined terciles. Perfect forecasts have values of 1 for all three skill metrics. Climatological forecasts (probabilistic forecasts that in our case each year predict a 1/3 chance of a high or low anomaly occurring) lead to values of 0 for R, 0.5 for the ROC area and 0 for the RPSS. In the computation of significance of the RPSS, sampling errors, i.e. the limited number of ensemble members, constitute a problem. They cause a bias in the RPSS when climatology is used as reference (Mason and Stephenson, 2008). Therefore, the reference for the calculation of the RPSS was generated by sampling randomly from the multinomial distribution with $p = (1/3, 1/3, 1/3)$ and $N = 15$ (the number of ensemble members). In the present paper each metric is designated as significant for p-values less than 0.05. For a data set of 30 years, this implies R is significant for values $> 0.31$, ROC area for values $> 0.69$ and RPSS for values that vary depending on the outcome of the random draw for the reference. We checked these procedures to determine significance by analysing hindcasts that have no skill. Such hindcasts indeed produced for all metrics a fraction of cells with significant skill near the expected value of 0.05 (the p-value), indicating that the procedures are correct.

To a large extent, we found that our results and conclusions, in terms of spatio temporal patterns of skill, are independent of the chosen metric. Hence, and because among the three metrics the correlation coefficient is the easiest to understand, we will discuss results mostly in terms of the correlation coefficient, which is in line with Doblas-Reyes et al. (2013). The sensitivity to the chosen metric and significant differences between these metrics will be discussed in Sect. 4.2.

All metrics were computed using the low and high level R packages "SpecsVerification" (Siegert et al., 2014) and "easyVerification" (Bhend et al., 2016), respectively. Metrics cannot be computed (because they become ill-defined) if observations or hindcasts within the entire 30 year period consist for more than one third of zeros or one sixth of

ties (i.e. equal values). Such skill gaps (i.e. the white terrestrial cells in Fig. 2 and 3) mainly occur in the far North due to rivers that are frozen for at least a month in winter.

**3        Results**

 **3.1        Spatiotemporal variation of skill in runoff forecasts**

Eighty-four maps of the skill of the runoff hindcasts were produced for all 12 initialisation months and all 7 lead months (all are presented in supplementary material Fig. S1). Two cross-cuts through that collection are shown in Fig. 2 (for a single initialisation month) and 3 (for a single lead month). The seven panels of Fig. 2 show the skill of the hindcasts initialised on April 1 as a function of lead time. Cells with an insignificant amount of skill are tinted yellow; cells where no metric could be computed remain white. In lead month 0, significant skill is found across almost the entire domain (99% of the cells). After the first lead month, the fraction of cells with significant skill gradually decreases to reach 16% at the longest lead time (lead month 6). This is more than expected for the case of completely unskilful simulations (5% of the cells), so at the end of the hindcast simulations significant skill that does not occur due to chance is still present in some regions. The general impression is that the pattern of skill does not move in space but that skill is fading, i.e. for individual grid cells R is mostly decreasing with increasing lead time. The same holds for initialisation in other months (see Fig. S1 in the supplementary material), with important exceptions better identified with Fig. 5 and discussed there.

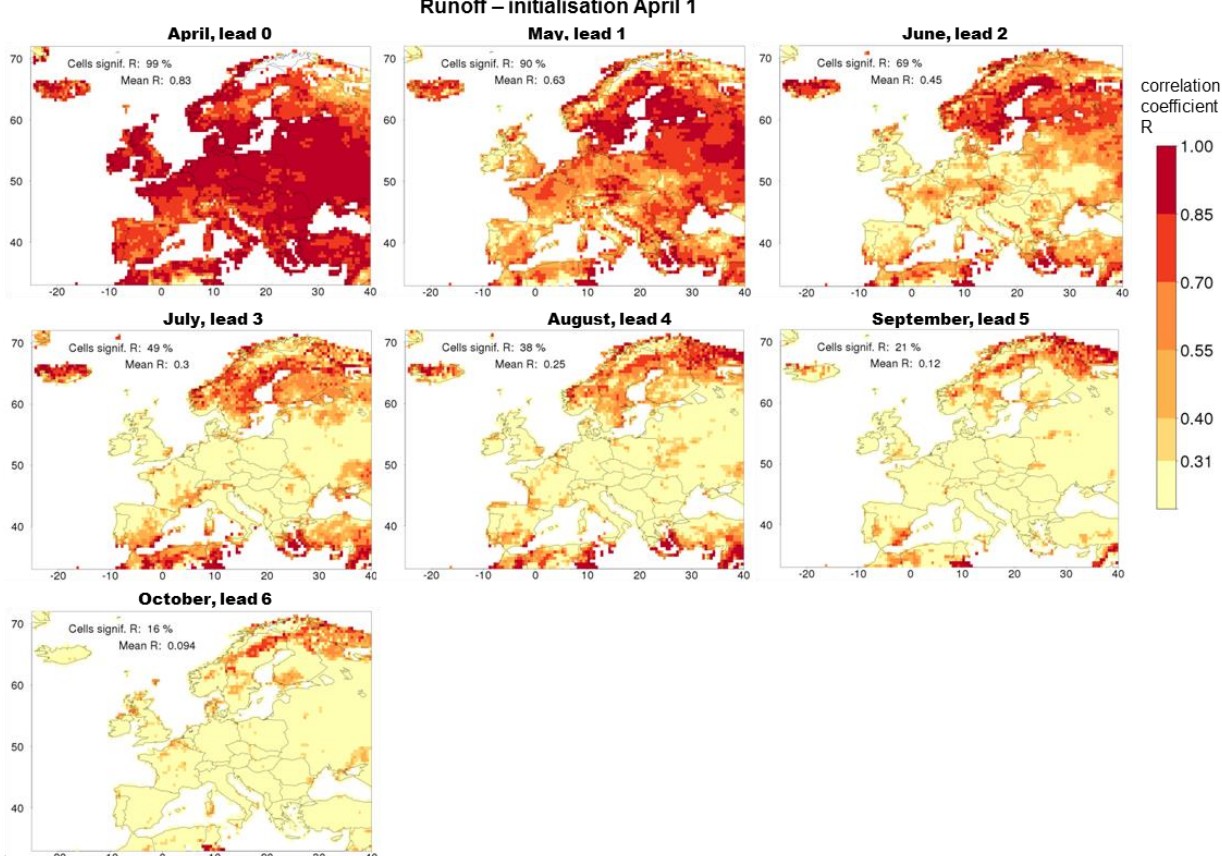

455

Figure 2:  The skill of the runoff hindcasts initialised on April 1 for all seven lead
457           months. The skill is measured in terms of the Pearson correlation coefficient
between the median of the hindcasts and the observations (R). The threshold
of significant skill lies at 0.31, so yellow cells have insignificant skill while
darker cells have significant skill. White, terrestrial cells correspond to cells
where observations or hindcasts consist for more than one third of zeros or
one sixth of ties. The legend provides the fraction of cells with significant
values of R (at the 5% level) and the domain-averaged value of R.



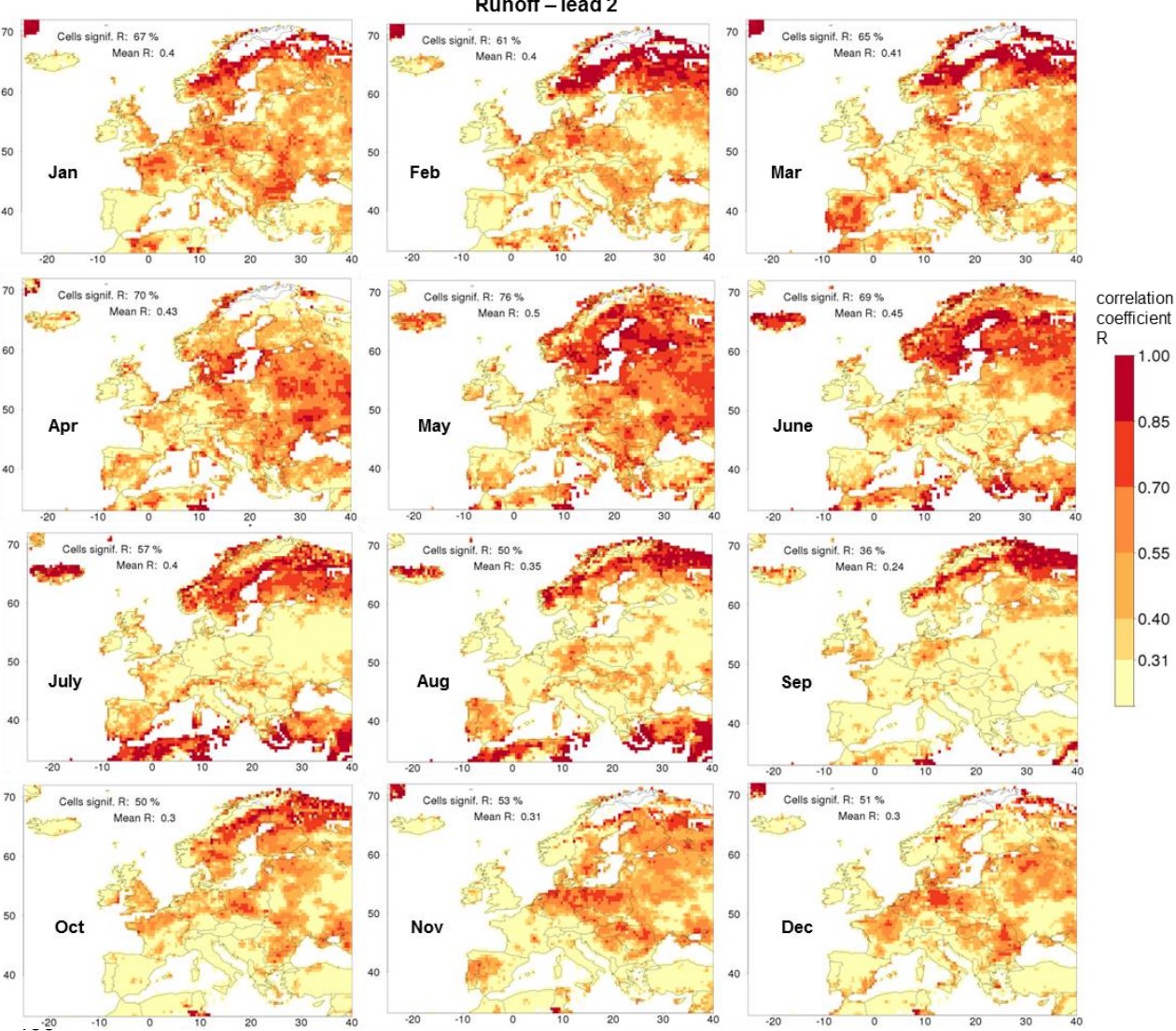


Figure 3:    Annual cycle of skill (R) of runoff hindcasts for 12 target months, initialised
at the beginning of the second month before (lead month 2). More
explanation is given in the caption of Fig. 2.



The twelve panels of Fig. 3 show the annual cycle of the skill of the hindcasts for lead
month 2, which is selected (also in Figures 6, 7 and 9) because at that lead time
approximately 50% of the cells have significant skill. Consistent skill (persistent during
at least 2 consecutive target months) is found in (causes of skill are reproduced here
from the companion paper, Greuell et al., 2016):
-    Fennoscandia. Much skill is present during the entire year, except for target months
November and December, and there is a dip in the skill in April. Most of the skill
is due to initial conditions of soil moisture. On average across the entire region, the
skill reaches a maximum in May and June, i.e. at the end of the melting season,
which is, as shown in the companion paper, largely due to initialising snow.
Compared to the rest of the peninsula, there is generally less skill along the

Scandinavian Mountain range. The companion paper shows some evidence that this
may be due to high variability of orographic rain, ill-represented in the S4 hindcasts.
- Poland and northern Germany. The core period lasts from November to January,
but it is extended with periods of less skill into October and the months from
February to May. Here the initialisation of soil moisture is the dominant cause of
skill. Snow initialisation contributes in April and May.
- Western France, more or less from Paris to Brittany and roughly from December to
May. Skill derives from the initialisation of soil moisture.
- The eastern side of Great Britain from January to April. Also here the skill derives
from soil moisture initialisation.
- Romania and Bulgaria. The core as well as the whole period are the same as that
for Poland and northern Germany.
- The southern part of the Mediterranean region from June to August. The high
amounts of skill are limited to the coastal parts of northern Africa, Sicily, southern
Greece, Turkey, Syria and Lebanon. This skill is due to initialisation of soil
moisture.
- The Iberian peninsula in March and August with smaller amounts of skill in months
in between. The skill derives mainly from soil moisture in the initialisation. In
March there is a minor contribution from skill in the forecasts of precipitation.
From Fig. S1 we broadly conclude that regions with skill for lead month 2 retain their
skill for other (longer) lead times, but that the magnitude of skill decreases with
increasing lead time as demonstrated in Fig. 2 (keep in mind that a change in lead time
corresponds to a change in target time by the same amount). To give an example: for
lead month 3 patterns in the skill maps look similar to those provided in Fig. 3 but
colours are fainter and target months shift by one month ahead. There are many
exceptions to this general rule, e.g. skill due to snow melt that suddenly appears at the
end of the melt season at longer lead times while it was not present during the lead
months before (see Fig. 5 and the companion paper). A more detailed regional analysis
of some of these features is left for future case studies.


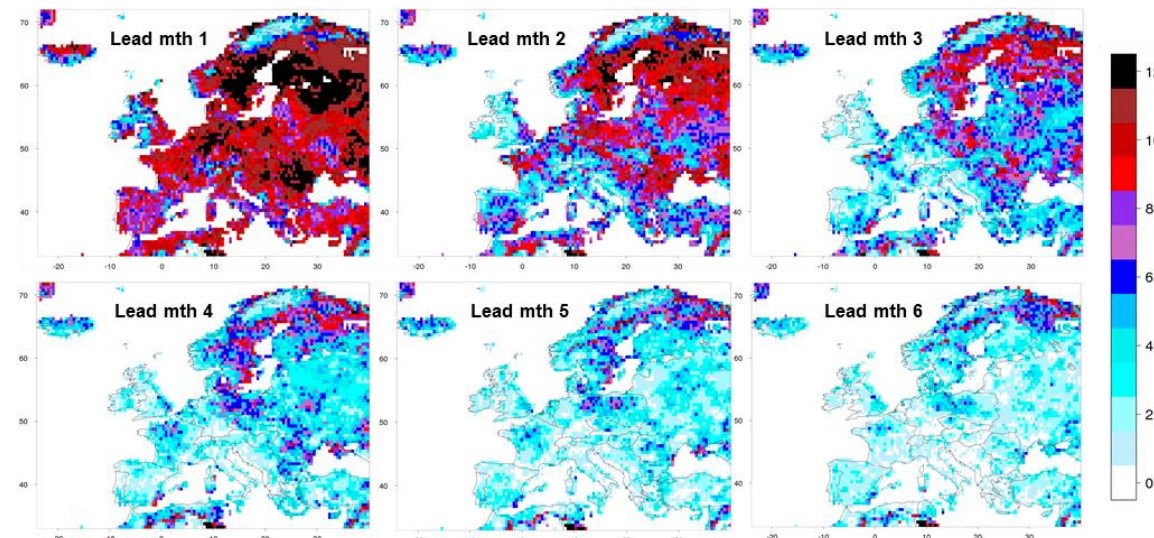

Figure 4:     Number of months in a year with significant skill (R) in the runoff
forecasts of lead months 1-6.


Figure 4 displays a synthesis in the form of a six maps with the number of the 12 months
of the year with significant skill for lead months from 1 to 6. In accordance with what
was also illustrated by Fig. 2, the amount of significant skill degrades with increasing
lead time. There is generally more skill all over the year towards the north and the
northeast. Many of the regions with very little or no skill are coastal regions (e.g.
northern coast of Spain), especially coastal regions on the western side of land masses
(e.g. western coasts of Denmark, southern Norway, Italy, Croatia and the British Isles),
and mountain regions (e.g. the Alps except for its southern fringe, mountains in northern
Norway and Sweden and the Tatra on the border of Poland and Slovakia). The British
Isles exhibit little skill, except for the eastern coast of Great Britain in late winter and
early spring (JFMA). Many of the regions that were listed before as having consistent
skill for lead month 2 also appear as foci of skill during the whole year, namely
Fennoscandia, northeast Germany and northwest Poland, Romania and Bulgaria,
Western France and the eastern side of Great Britain. The companion paper shows that
for regions with skill during a large part of the year, this skill is due to initial conditions
of snow and/or soil moisture.

These pan-European results can be compared to those of Bierkens and Van Beek (2009).
They found maxima in predictability of winter discharge in Northern Sweden, Finland,
the region between Moscow and the Baltic Sea, Romania and Bulgaria, and Eastern
Spain. For the winter there is crude agreement with the current study about Northern
Sweden, Romania and Bulgaria, but not about the other regions. For the summer,
Bierkens and Van Beek (2009) compute maxima in skill for Southern Spain, Sardinia,
Western Turkey and South-western Finland, a pattern that broadly agrees with the
locations of the summertime maxima in skill we find (most of Fennoscandia and
southern part of the Mediterranean region).

Singla et al. (2012) found considerable skill in the Seine basin for low flows from June
– September, a bit more eastern from the region where we found skill. Trigo et al. (2004)
using a statistical model based on December NAO indices found skill for JFM discharge
(and hydropower production) for the Douro, Tejo and Guadiana basins covering most
of central and western Iberia. We confirm this skill for March in these regions, but not
for January and February while we find some skill for later months (March until
August). Svensson et al. (2015) using a statistical model, based on NAO indices and
river flow persistence, found good skill for winter river flows on the eastern side of the
British Isles, consistent with our findings, and low but just significant skill along its
western coast, which we do not reproduce.

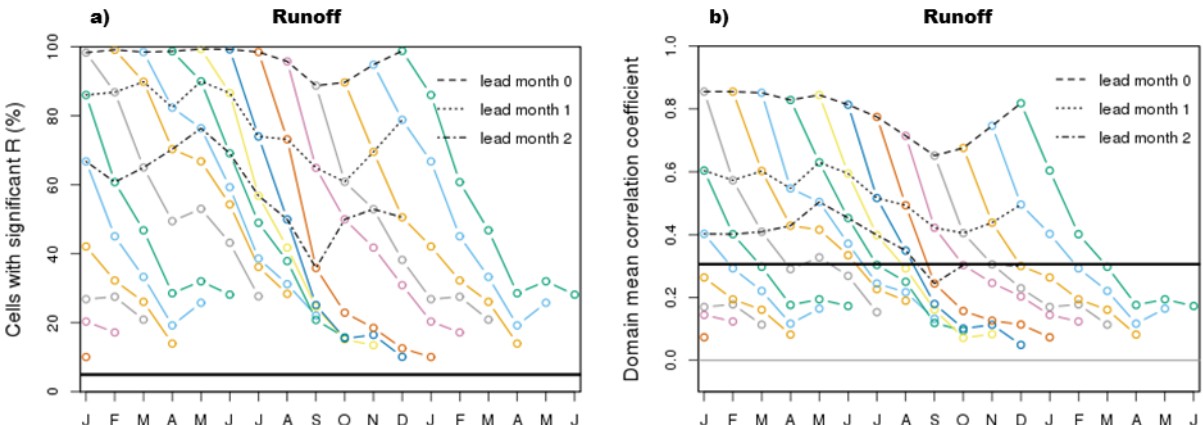


Figure 5:  a) Fraction of cells with significant skill (in terms of R), and b) domain
average correlation in the runoff hindcasts, as a function of initialisation
month and lead time. Each coloured curve corresponds to the hindcasts
initialised in a single month. For better visualisation, parts of the curves that
end in the next year are shown twice, namely at the left-hand and the right-
hand side of the graph. Black lines (dashed, dotted and dashed-dotted)
connect the results for identical lead times. The horizontal line in a) shows
the expected fraction of cells with significant skill, in the case that the
hindcasts have no skill at all (5%), and in b) the minimal magnitude of the
correlation of a single cell for it to be statistically significant.


Fig. 5a summarizes skill across the domain in terms of the fraction of cells with
significant R for all initialisation and lead months. Overall there is a considerable
amount of significant skill, with a minimum roughly from August to November and a
maximum in May. For lead month 2 the fraction of cells with significant skill varies
between 36% (September) and 76% (May). In all of the 84 combinations of initialisation
and lead month, the theoretical value of no skill at all (5%) is exceeded, implying that
there are (small) pockets of skill even at lead month six. Individual curves show that
skill is lost with increasing lead time. The exception is formed by hindcasts starting in
November, December and January which gain skill when they progress from April to
May, a phenomenon caused by initial conditions of snow that takes longer or shorter to
completely melt in (late) spring. For details, see the companion paper. Fig. 5b shows
decay and gain trends of the domain-averaged R. It shows that a forecast initialised in
February exhibits higher domain average skill into June (5 lead months) than one
starting in July into September (2 lead months). Similar summary plots for the other
skill metrics are presented in the Fig. S2 and discussed in Sect. 3.4.

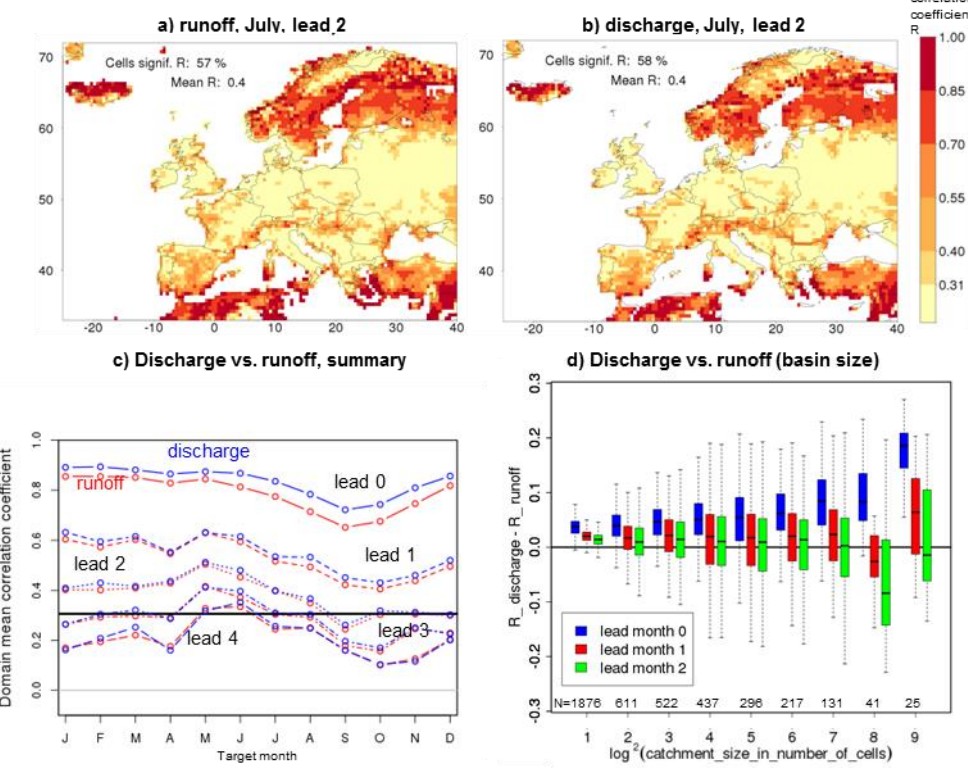

Figure 6:    Comparison of the performance of the hindcasts of discharge and runoff
using the pseudo-observations for verification. The two maps display R for
runoff (a) and discharge (b) for hindcasts initialised on May 1 and target
594                 month July (see further explanation in Fig. 1). Panel c depicts the annual
cycle of the domain-averaged R for runoff (red) and discharge (blue) for
lead months 0 to 4. The horizontal line at 0.31 is the threshold of significance
for a single cell. Panel d is a box plot of the difference between R for
discharge and runoff as a function of the basin size. Each bin $i$ contains the
results for all basins with a maximum of $2^i$ cells and more than $2^{(i-1)}$ cells,
e.g. bin 4 is for all basins with a size from 10 to 16 cells. Boxes represent
the interquartile range and the median; whiskers extend to minimum and
maximum values found in the bin. All values are average differences over
the twelve months of the year and results are shown for three different lead
604                 times. The values above the abscissa give the number of cells in each bin.

## 3.2 Spatiotemporal variation of skill in discharge forecasts

This sub-section compares the skill for discharge with the skill for runoff. The two maps of Fig. 6, which depict the skill in runoff and discharge hindcasts for July as lead month 2, show a high degree of similarity in terms of the patterns and the magnitude of the skill. The same holds for other target months and lead times (not shown). There are, however, subtle differences because rivers aggregate the skill, or lack of skill, from the whole upstream part of their basin. As a result, cells containing rivers with large basins may contrast against adjacent cells if these contain rivers with a small, local basin. Indeed, some downstream parts of large rivers stick out in the skill map for discharge, but not in the skill map for runoff. An example in Fig. 6b are the reaches of the Danube along the Romanian-Bulgarian border, which show more skill than local small rivers in adjacent cells, because some upstream parts of the Danube have more skill than the region around the Romanian-Bulgarian border. An example that demonstrates the opposite is the downstream part of the Loire showing less skill than local small rivers, because upstream parts of the Loire have less skill than small, local rivers in the downstream part.

Domain summary statistics of skill also differ slightly between runoff and discharge. Figure 6c compares the annual cycle of the skill in discharge with the skill in runoff at five different lead times. Here we show the difference in the domain-averaged R instead of the fraction of cells with a significant R because in lead month 0 that fraction is close to one for both variables. In terms of the domain-averaged R, predictability is higher for discharge than for runoff for the first lead month. On average over the 12 months of the year, the difference is 0.049. We ascribe this result to the combined effect of the delay between runoff and discharge, with variations in discharge being later in time than the corresponding variations in runoff, and the general tendency of decreasing skill with lead time. The curves for the different lead times in Fig. 6c show that the difference in skill between the two variables gradually disappears with increasing lead time (an annual average of 0.020 and 0.012 for lead months 1 and 2, respectively). This is compatible with the given explanation for the difference and the fact that the rate with which skill is lost gradually decreases with increasing lead time.

We finally analysed whether the difference in skill between discharge and runoff was a function of the size of the basin (Fig. 6d). For the first lead month, when on average there is more skill in discharge than in runoff, the difference increases with the size of the basin. Again, this can be explained by the combination of the skill decaying with time and the delay between runoff and discharge, with the delay increasing with the size of the basin. For longer lead times (from lead month 1 on), when the domain-averaged difference in skill has become very small, the figure shows no effect of the basin size. Referring to the comparison between runoff and discharge in panels Fig. 6a and 6b for

lead month 2, cases like the Danube (more skill than local rivers) and the Loire (less
skill than local rivers) tend to cancel when the entire domain and year are considered.


**3.3     Verification of discharge with pseudo- and real observations**

So far, all skill was determined by using the discharge generated with the reference
simulation. i.e. with pseudo-observations. In this section, this "theoretical skill" will be
compared with the skill determined with real discharge as observed at gauging stations
("actual skill") from the GRDC and EWA databases. Fig. 7 compares the theoretical
skill (Fig. 7b and 7d for large and small basins, respectively) with actual skill (Fig. 7c
and 7e for large and small basins, respectively) for a single combination of a target
month (May) with a lead month (2). Small basins are defined as smaller than one 0.5° x
0.5° grid cell, large basins are larger than 9900 km² (see Sect. 2.2).

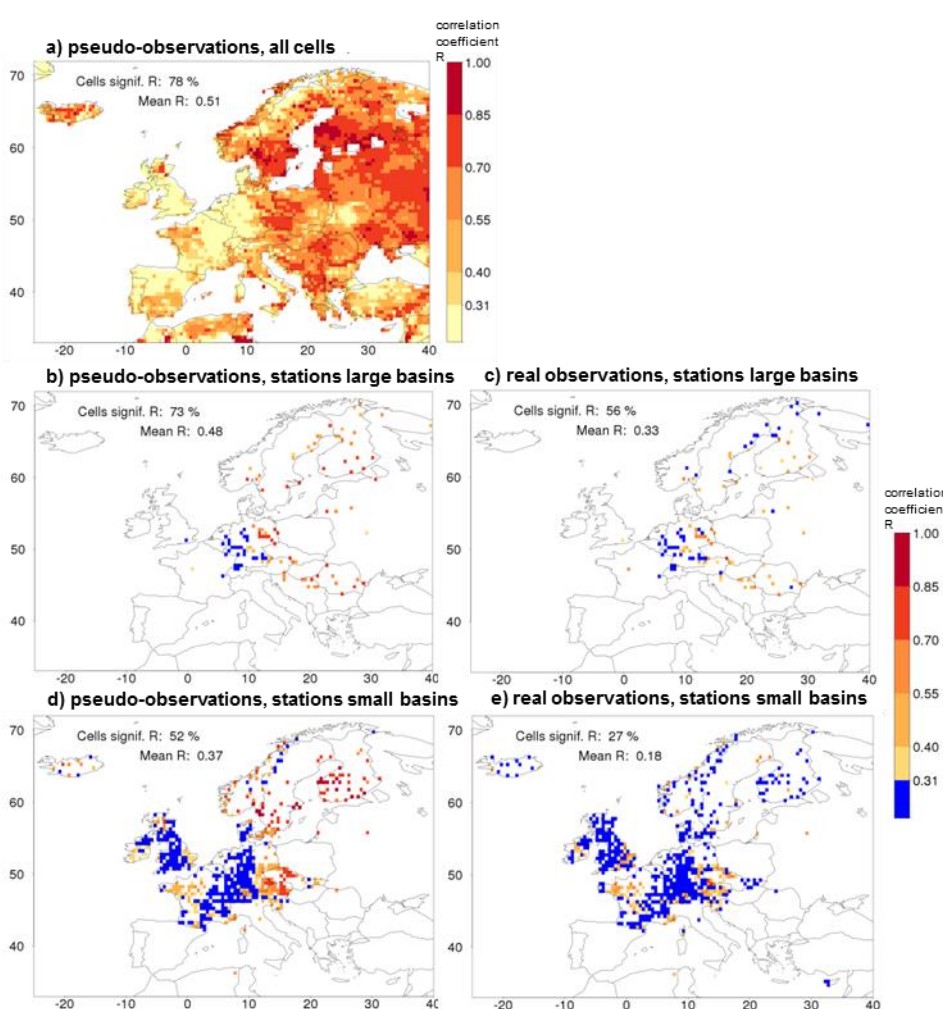


Figure 7:    Skill (R) of the discharge hindcasts for May as lead month 2 (initialisation
on March 1). In sequence: a) discharge verified with pseudo-observations,
b) as a) but for cells with real observations representing large basins only,

c) discharge verified with real observations for large basins. Panels d) and e) are identical to b) and c), respectively, but for cells with real observations representing small basins. More explanation is given in the caption of Fig. 1 but in panels d) and e) cells with insignificant skill are coloured blue instead of yellow for better contrast.

For this combination of May forecasts initialised in March, a substantial degradation in skill is found when the pseudo-observations are replaced by real observations. In terms of the fraction of cells with significant skill, the reduction is from 73 to 56 % for large basins and from 52 to 27 % for small basins and the domain-averaged R decreases from 0.48 to 0.33 for large basins and from 0.37 to 0.18 for small basins. Especially the basins in northern Fennoscandia lose much of their skill when using actual instead of pseudo-observations. In this region VIC also performed poorly in reproducing historic flows. Specific discharge was underestimated and the annual cycle was poorly reproduced; especially the spring peak occurred too late and lasted too long (Greuell et al., 2015). In central Europe useful skill remains when using real observations, both for small and large basins. This is a region where VIC reproduced well annual cycles, though interannual variations in low flows were overestimated. For a few stations in Northwest France and Southeast England actual skill is larger than theoretical skill.

Fig. 8 compares actual with theoretical skill for all target months and two lead times by considering the domain-mean R. Similar figures for the other skill metrics are presented in Fig. S4 and discussed in the next section. The reduction of actual relative to theoretical skill occurs for all combinations of target and lead months and does not exhibit a clear annual cycle. On average across all target months and for lead month 2, the ratio of actual to theoretical skill is 0.667 (0.258 divided by 0.387) for large basins and 0.538 (0.156 divided by 0.290) for small basins. This is comparable to Van Dijk et al. (2013), who found a ratio of actual to theoretical skill of 0.54 for 6192 basins worldwide in terms of the ranked correlation coefficient.

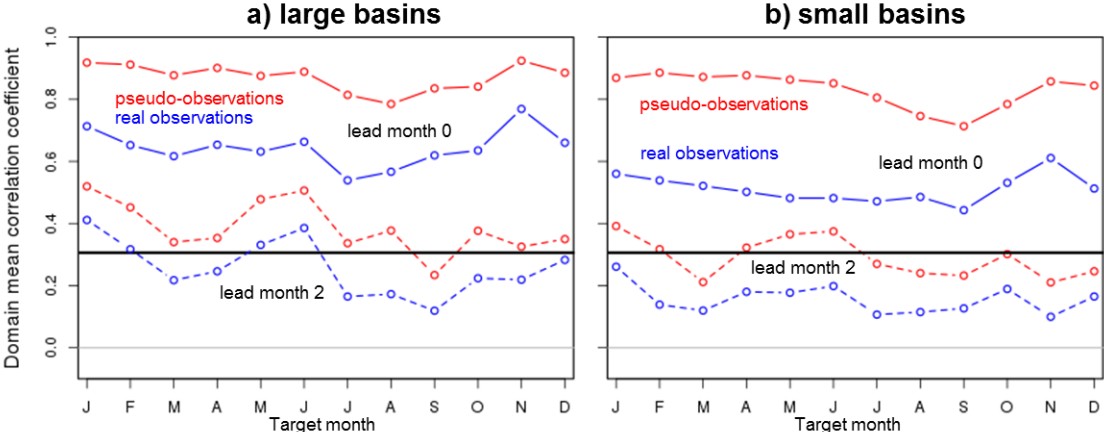

Figure 8: Comparison between verification of discharge with pseudo- (red) and real (blue) observations in terms of the annual cycle of the domain mean R. The horizontal line at 0.31 is the threshold of significance for a single cell. Results are shown for cells representing large basins (left) and cells representing small basins (right). Both panels depict cycles for lead months 0 and 2 only.

Comparing skill for small basins with skill for large basins in Fig. 8, we notice two differences. Firstly, in terms of the domain mean R, theoretical skill is higher for large basins than for small basins (0.39 and 0.29, respectively, for the annual mean and lead month 2). However, this result holds for the cells with observations. If all cells of the domain are considered, this difference becomes insignificantly small. So, the apparent difference in theoretical skill between large and small basins can be attributed almost entirely to the geographical distribution of the discharge monitoring stations, with stations on small basins being relatively more often located in regions with relatively little skill like Germany, France and the British Isles than large basin stations.

The second effect of the size of basins is that the ratio between theoretical and actual skill is larger for small basins than for large basins, at least for lead month 0. This is perhaps even more clear from Fig. S3 in the supplementary material. We speculate that this is due to a combination of two effects. Firstly, there is more skill in simulations of historic streamflow in large basins than in small basins (Van Dijk and Warren, 2010, confirmed for VIC in Europe by Greuell et al. 2015). Secondly, as Van Dijk et al. (2013) demonstrated, the ratio of actual to theoretical skill in the hindcasts is almost linearly related to the skill of simulating historic streamflow. Combining these two relationships confirms the relationship that we found, namely an increase in the ratio of actual to theoretical skill with basin size.

Finally, we investigated to what extent these results are affected by human interference, keeping in mind that the simulations are naturalized, while the observations include human impacts to a variable but unknown degree. Human interference is expected to have a negative effect on actual skill and hence on the ratio of actual to theoretical skill. For relatively natural "large basins" (AAPFD < 0.3; see end of Sect. 2.2), the ratio of actual to theoretical skill was computed in terms of the domain mean R, averaged across all target months and for lead month 2. We found a ratio of 0.686, which should be compared to a ratio of 0.667 for the entire set of large basins (see above). So, as expected the ratio is larger for basins with less impact. However, since the difference between the two ratios is small we conclude that the effect of evaluating naturalised runs against observations that are obviously affected by human interference, contributes only little to the difference between actual and theoretical skill. A similar analysis was not applied to the collection of small basins with observations, since these are smaller than the spatial resolution of the simulations.

744

## 3.4 Results for other skill metrics

746

So far, skill was measured in terms of the correlation coefficient between the median of the hindcasts and the observations (R) only. This section compares those results, for runoff, with results in terms of other skill metrics. Fig. 9 gives an example for one particular target month and lead month, i.e. target May initialised in March (lead 2). Fig. 9a, 9b and 9c show the skill patterns for R, for the ROC area for Below Normal (BN) years and for the RPSS. The three patterns are spatially similar to a large degree, though the magnitudes and number of significant cells do differ. The pattern of the map of the ROC area for Above Normal (AN) years (see Fig. S1) is also similar to the patterns of the three maps shown. On average across all lead and target months, among cells that have significant R, 89% and 84% also have a significant ROC score for the BN tercile and the AN tercile, respectively, and 65% also have significant RPSS scores. The fraction of cells with no significant R, but with significant ROC or RPSS remains below the 5% level across all target and lead months.

760

The agreement that we find between the patterns of the different metrics is in accordance with a result mentioned in a global analysis of seasonal streamflow predictions by Van Dijk et al. (2013) who found high spatial correlation between the different skill metrics they used (among which R, the RPSS and the ranked correlation coefficient).

765

766

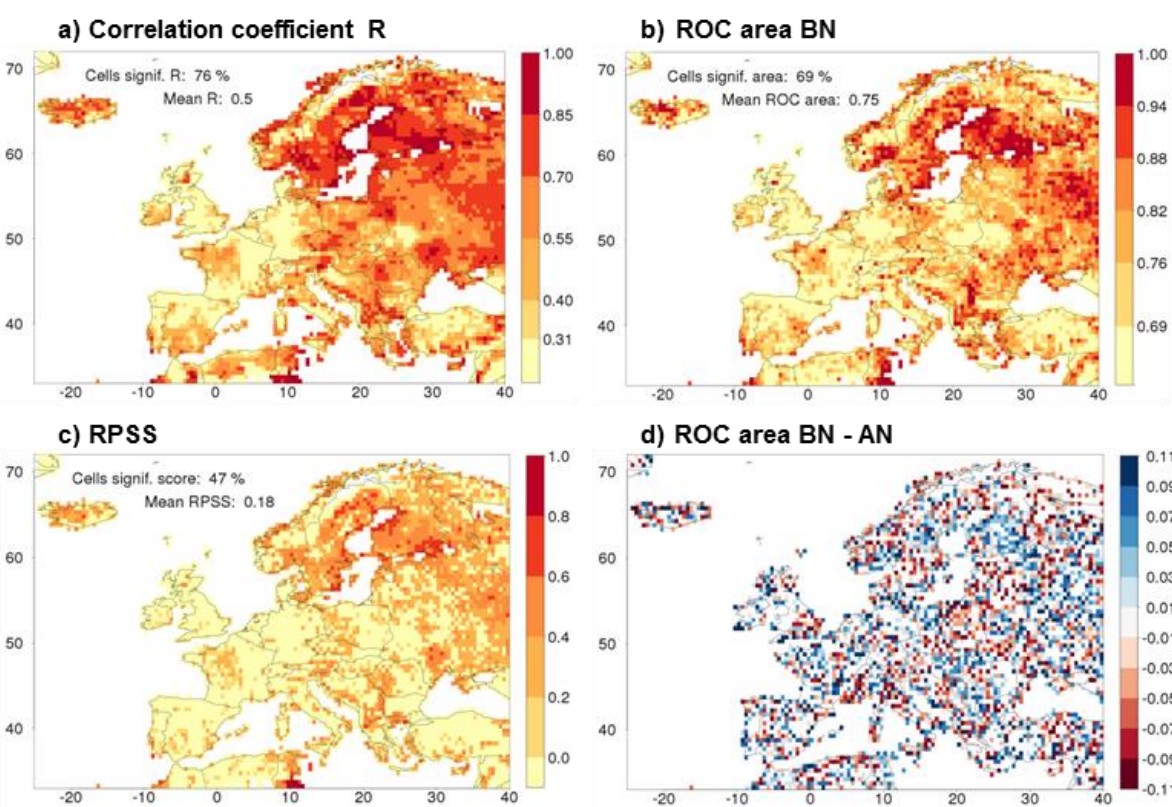

767

Figure 9:    Maps of different skill metrics for one combination of a target month (May)
and a lead month (2) of the runoff hindcasts. Panels show a) R, b) the ROC
area for the below normal tercile, c) the Ranked Probability Skill Score
(RPSS) and d) the difference in ROC area between the BN and AN terciles.
In panels a), b) and c) skill is not significant in cells with a yellow colour.
Legends provide the fraction of cells with significant values of the metric
and the domain-averaged value of the metric.

Although the different nature of the different metrics makes the interpretation of quantitative differences between them difficult, this is not true for the ROC areas for the different terciles. For the particular combination of May as target month and lead month 2 shown in Fig. 9, the domain-mean ROC area is largest for the BN tercile (0.75), slightly smaller for the AN tercile (0.73) and much lower for the near-normal (NN) tercile (0.58, see Fig. S2c and d). A similar tendency is found in the fraction of cells with a significant ROC area (69%, 63% and 21%, respectively). In fact, in all combinations of lead and target month the fraction of significant cells is larger for the BN than for the AN tercile, as shown in Fig. 10. However, the AN and BN fractions of cells tend to become equal (i) when they approach 1.0, (ii) when they approach the limit of no skill (5%) and (iii) during target months from October to January. Finally, Fig. 9d presents a map of the difference between the BN and the AN ROC area for May as lead month 2. There is some organisation in the pattern but regions with a positive or a negative difference between the two tend to be smaller than the regions with significant skill in the maps of e.g. Figs. 2 and 3. Also, we did not detect much consistency, in the sense of persistence during at least two consecutive target or lead months, in the patterns of the difference between ROC AN and ROC BN.

In Fig. 9c the fraction of cells with a significant value of the RPSS is 47%, which is somewhere between the fractions for ROC areas of the three terciles because the RPSS represents the skill across all terciles.

For other combinations of target and lead months the results of this analysis are qualitatively similar, see supplementary figures. All metrics show a minimum value in the annual cycles in either September or in October, irrespective of lead time; maxima are attained in February for lead month 0 shifting to May at longer lead times (Fig. S2). We would finally like to note that, while in this sub-section we discussed runoff, we made similar figures and calculations for discharge. Results for these two variables are almost identical.

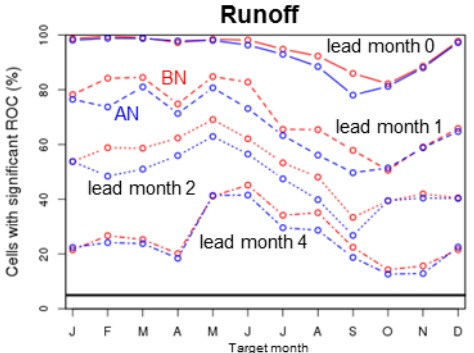

Figure 10: Skill of the runoff hindcasts in the Below Normal (BN) compared to the skill of the runoff hindcasts in the Above Normal (AN) tercile. The plot depicts annual cycles of the fraction of cells with a significant ROC area for the two terciles and for four lead months.

## 4    Discussion

### 4.1    Theoretical versus actual skill

In the analysis of the differences between theoretical and actual skill, two essential questions are: a) What are the conceptual differences between the physical systems that generate the pseudo- and the real discharge observations, i.e. between the model reference run and the real world. To answer this question, the components in the upper and the lower box of the diagram in Fig. 1 need to be compared. b) What are the expected effects of these differences on skill, i.e. on the comparison with the hindcasts. To answer this question, the components that differ between the real world and the model reference run need to be compared with the model hindcasts. The rule then is that skill decreases with increasing disagreement between a component of the hindcast system and the corresponding component of one of the other systems. The following components differ between the real world and the model reference simulation:

1. Real meteorology differs from the meteorology assumed in the reference simulation (WFDEI), both during the spin up period and during the hindcast period. During spin up, model reference run and hindcasts have identical meteorological forcing (WFDEI), which differs from real meteorology. Therefore, this difference is expected to lead to more theoretical than actual skill. During the hindcast period, all three systems have different meteorological forcing. We do not have a well-founded expectation about any biases between these three forcings and, hence, we have no expectation about its effect on the difference between theoretical and actual skill. However, in Europe and beyond the first lead month almost all skill in the seasonal forecasts is due to the initial conditions (see the companion paper). Therefore, beyond the first lead month and in Europe differences in forcing during the hindcast period have a negligible effect on skill.

2. Models are imperfect, in terms of physics and in terms of spatial and temporal discretisation, so model hydrology differs from real world hydrology. Hindcasts and the pseudo-observations are produced with the same model, so imperfections in model hydrology are expected to lead to more theoretical than actual skill. One assumption implicitly made in the diagram is that the basin of the observation station and the model basin are identical. This is not the case (see Sect. 2.2), so differences between observation and model basin form an additional cause of disagreements between theoretical and actual skill. Again, this will favour theoretical skill with respect to actual skill since basins are identical in the hindcasts and the reference simulation. In particular, differences in meteorological forcing between the basin of the observation station and the model basin might reduce actual skill. Van Dijk et al. (2013) investigated this aspect by making simulations for Australia at different spatial resolutions and verifying with networks of observations with different spatial densities. They found that the resolution and perhaps the quality of the forcing data contributed at least half to the difference between theoretical and actual skill.

3. In the real world discharge observations are subject to measurement errors. Measurement errors of discharge are not constant over time (due to varying cross sectional areas, following erosion and sedimentation) and therefore add noise to the data; noise always reduces skill. There is no equivalent of this error in the model environment. Hence, as for differences 1) and 2) this difference is expected to lead to more theoretical than to actual skill.

4. Initial conditions are absent in this list of differences since they are not independent components but entirely determined by two components of the system listed above, namely meteorology and hydrology. Alternatively, initial hydrological conditions could be taken from observations or by assimilation of observations into model calculations. In that case, initial conditions would become an independent or semi-dependent component of the system. However, while model initial conditions would, of course, differ from real initial conditions, the two model systems had identical initial conditions. Hence, this difference would again be expected to lead to more theoretical than to actual skill.

In summary, all of the conceptual differences between the generation of pseudo- and real observations are expected to lead to more theoretical skill than actual skill, except for the difference in meteorology during the hindcast period, which has, in the case of Europe beyond the first lead month, a neutral effect, and otherwise an unknown effect.

Our data analysis, Sect. 3.3, broadly confirms that theoretical skill exceeds actual skill. We also found cases where actual skill exceeds theoretical skill beyond lead month 0, like for a few stations in Northwest France and Southeast England in Fig. 7. We ascribe such cases to chance.

It is interesting to discuss what would happen in the utopian case that the system of the
model reference run would converge with the real world, i.e. if model meteorological
forcing and hydrology would approach perfection and if measurement errors would
approach zero. Equality of the two systems would, according to the analysis above, lead
to equality of theoretical and actual skill. However, we like to note that at the same time
optimisation of the model system can lead to a degradation of the theoretical skill due
to unrealistic memory time scales in the storage compartments of the hydrological
model before optimisation. If this memory, from stored water in either snow, soil or
aquifer is too strong, then skill will reduce with calibrating the model towards more
realistic storage accumulation. However, if this memory is too small before improving
the model, then, of course, the reverse would happen and skill increases with
optimisation.
An example proving this statement is a model that accumulates too much snow. The
model will do so both in the initial state of the reference simulation and the initial state
of the hindcasts and since more snow leads, at some stage of the melting season, to more
predictive skill, theoretical skill will be overestimated. A perfect model, accumulating
less but more realistic amounts of snow, would exhibit less skill. Another example is
predictive skill caused by interannual variations in the initial amount of soil moisture
and/or groundwater. A model that is imperfect because it overestimates the transport
speed of water through the soil and the groundwater reservoirs will do so both in the
reference simulation and the hindcasts. Predictive skill due to soil moisture initial
conditions will then occur too early. Compared to the model that overestimates transport
speed, a perfect model with smaller, realistic transport speed would yield less theoretical
skill at the early lead times.
Hence, theoretical skill is not equal to the maximum that could be accomplished if
hydrological model and meteorological forcing during the reference simulation were
perfect.
The version of VIC used in this study was calibrated by Nijssen et al. (2001) in a crude
way, in the sense that they assumed no spatial variation of the parameters set by
calibration within almost the entire European continent. Improving the calibration of
VIC would be an obvious candidate for trying to improve the seasonal predictions
discussed in this paper. This should lead to higher actual skill. However, the two
examples discussed above show that theoretical skill may actually, for certain locations,
months of initialisation and lead months, decline due to the recalibration.

**4.2    Results and uncertainties**


There seems to be a broad correspondence between the probabilistic forecast
verification presented here and the model validation presented in Greuell et al. (2015)
and Roudier et al. (2016). These studies found that average discharge and inter-annual
variations therein are well reproduced against observations, consistent with our result

that in the first lead month all skill scores, also against real observations (see Fig. S4 for the lead 0 results), are good for large parts of Europe.

However, the relation between a model's ability to simulate historic streamflow and its ability to generate skill in seasonal forecasts is complex. There is, for instance, no reason to expect that regions with more theoretical skill than other regions would generally correspond to regions with better historic streamflow simulations. Large model stores of soil moisture and snow tend to lead to more theoretical skill, whether these stores are realistic or not. If they are not realistic, simulations of historic streamflow will be poor, despite the forecast skill. Another example of the problematic relation between validation and verification is that, even in perfect models, regions with small model stores of soil moisture and snow and regions with large interannual variation in precipitation will exhibit small amounts of theoretical and actual skill. So, regions with high quality historic streamflow simulations may for good reasons have little skill in the forecasts.

However, what we would expect is that regions of poor model performance have little actual skill (not necessarily little theoretical skill) in the forecasts. In our work, this statement is broadly confirmed by the basins in northern Fennoscandia, which lose much of their skill when using actual instead of pseudo-observations (Fig. 7). In this region VIC indeed performed poorly in reproducing historic flows. Good model performance probably is a necessary (but not sufficient) condition for the generation of actual skill in seasonal forecasts. This is exemplified by some regions with considerable amounts of actual skill in central Europe (e.g. northern part of the Balkans and the Elbe basin in Fig. 7), where VIC's simulations of historic streamflow are much better than in northern Fennoscandia.

In a future extension of our work, an objective method like cluster analysis could reveal regions where skill has a similar signature. This could lead to an improved assessment of the physical and climatological factors that are responsible for the spatial variations in skill found in this and its companion paper.

There also seems to be a broad correspondence between the regions and seasons with skill identified in the present work and those identified in more spatially or temporally confined studies based on entirely different physical or even statistical models. Without repeating the more detailed description in the Introduction and the closer comparison in Sect. 3.1, we restate here that the results of Bierkens and van Beek, (2009) and Thober et al. (2015) were similar at the European domain. These pan-European studies, like ours, confirm more regional studies such as for the British Isles (Svensson et al., 2015) or France (Céron et al., 2010; Singla et al., 2012). Though a high resolution study like the latter may add much spatial detail, this does not change the region and season of skill.

Our results are based on a forcing with the 15 member, monthly initialized, 7 month forecast version of ECMWF System 4, basically because at the start of this work that hindcast was the only one accessible to us, but also because it allows verification at the monthly resolution. Alternatively, we could have used the 51 member seasonally initialised (4 times per year), 7 month forecast version of the same model. That would have provided us with better constrained, more precise statistics (larger sample size), or would have allowed assessment of more percentiles (e.g. quintiles instead of terciles) at similar precision. However, the variation of skill over a year would not have been resolved with such detail as in the present work. Finally, a 15 member, seasonally initialized, 12 month forecast version of System 4 is available. Our results show that for some regions at lead month 6 still a few, small pockets of consistent skill remain, suggesting that extending the forecast for our domain might be worth exploring.

Other seasonal forecasting systems, based on different coupled ocean-climate models, could have been used as meteorological forcing, such as CFSv2 (Saha et al., 2014) and Glosea5 (MacLachlan et al., 2014). Given that, at least at large scales, multi model ensembles exhibit better climate forecast skill than single models, it is interesting to investigate if that additional skill also propagates into river flow forecasts. While this seems to be true for the Eastern United States (Luo & Wood, 2008) it is not known if similar conclusions could be drawn for Europe. A similar reasoning can also be extended to the hydrological models: using a multi climate model ensemble to force a multi hydrological model ensemble might also provide improved skill, as the latter models may be complementary in the regions and seasons of best model performance. Bohn et al. (2010) showed some advantage of using an ensemble of three hydrological models (but with a single forcing), over using only the best of the three, but only after bias correcting the hydrological output and making a linear combination of them with monthly varying weights.

## 4.3    Implications and recommendations

Many conclusions drawn from this work are valid at the scale of our domain and not necessarily at the scale of river basins. Only in some parts of our analysis, especially where we focused on the annual cycle of the skill (Fig. 3), regional patterns at a scale smaller than that of the domain were discussed. This was done in a qualitative way.

For applications of these seasonal forecasts in decision making processes at (sub) basin level, a more detailed skill analysis is recommended for that specific (sub) basin, preferably after a better model calibration for that same basin. The facts presented in this study that anomaly correlations and ROC scores for the AN and BN terciles are significant for large parts of the domain for lead times up to several months, supported by (fairly) positive validation results for VIC (Greuell et al., 2015; Roudier et al. 2016), suggest these anomaly forecasts are good enough to be used as such. However, areas of significant RPSS are much smaller and remain significant for shorter lead times.

Spatially distributed calibration of VIC model parameters, or distribution based calibration of modelled discharge to observations, or both, might also increase the RPSS. This might then allow forecasting of absolute discharge magnitudes and thus inform decision making processes that involve certain absolute discharge thresholds.

In Sect. 3 (Results) we already discussed the probable reasons for skill, which are much elaborated on in the companion paper. In general that paper shows that for most areas skill in runoff is caused by initialising snow and /or soil moisture properly, only in few areas and seasons skill in precipitation or skill in temperature and evapotranspiration adds to that beyond the first lead month. This has two implications: one is that, if ever the skill of seasonal climate forecasts improves for Europe, this may well translate to improved seasonal river flow forecast too. The second is that better initial conditions of snow water equivalent and soil moisture from observations may do the same, but the latter only if the spatial distribution of the soil moisture storage capacity is more realistic too (see Sect. 4.1).

Overall the present analysis shows that especially in winter, spring and early summer, there is potentially good skill to forecast runoff and discharge in large parts of Europe, with considerable lead time. While this broadly confirms previously published work, the present study (while being specific to our model setup) gives much more spatial and temporal (season and lead time) details. As such it provides a good basis to support operational forecasts and to add information about skill to seasonal forecasts, which is very important for proper value assessment and decision making.

## 5    Conclusions

This paper is the first of two papers dealing with a model-based system built to produce seasonal hydrological forecasts (WUSHP: Wageningen University Seamless Hydrological Predictions). The present paper presents the development and the skill evaluation of the system for Europe, the companion paper provides an explanation of the skill or the lack thereof.

First, "theoretical skill" of the runoff hindcasts was determined using the output of the reference simulation as "pseudo-observations". Using the correlation coefficient (R) as metric, hot spots of significant skill were found in Fennoscandia (from January to October), the southern part of the Mediterranean (from June to August), Poland, northern Germany, Romania and Bulgaria (mainly from November to January) and western France (from December to May). There is very little or no significant skill all over the year in some coastal and mountainous regions. The entire British Isles exhibit very little skill, except for the eastern coast of Great Britain. If the entire domain is considered, the annual cycle of skill has a minimum roughly from August to November and a maximum in May.

Runoff and discharge show a high degree of similarity in terms of the spatial patterns and the magnitude of the skill. However, when averaged over the domain and the year, predictability is slightly higher for discharge than for runoff for the first lead month (by 0.049 in terms of R), but the difference decreases with increasing lead time. We also found that for lead month 0 the difference between discharge and runoff skill increases with the size of the basin.

Theoretical skill as determined with the pseudo-observations was compared to actual skill as determined with real discharge observations. On average across all target months and for lead month 2, skill reduction due to replacing pseudo- by real observations is larger for small basins than for large basins.

Spatio-temporal patterns for the different skill metrics considered in this study (correlation coefficient, ROC area and Ranked Probability Skill Score) are similar to a large degree. ROC areas tend to be slightly larger for the below normal than for the above normal tercile but not during target months from October to January.

## 6    Author Contributions

Greuell and Hutjes designed the experiments, with suggestions from the other co-authors. Franssen and Greuell developed the workflow scripts and performed all the simulations. Greuell and Franssen developed the analyses and plotting scripts in R. Biemans did the LPJmL work on AAPFD. All co-authors participated in repeated discussions on interpretations of results and suggested ways forward in the analysis. Greuell prepared the first version of the manuscript with contributions from all co-authors. Hutjes and Greuell prepared revisions of the manuscript with contributions from all co-authors.

## 7    Conflicting Interests
The authors declare that they have no conflict of interest.

## 8    Acknowledgments
This study was financially supported by the EUPORIAS project (EUropean Provision of Regional Impact Assessment on Seasonal-to-decadal timescale); grant agreement No. 308291, funded by the European Commission (EU) project in the Seventh Framework Programme. We thank the valued suggestions and insightful comments from two (anonymous) reviewers that contributed to an improved version of the manuscript.

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
