# Peer review of "with pseudo- and real observations"

_Hydrology and Earth System Sciences, 2016_

## Referee Comment (RC1) · Anonymous Referee #1 · 30 Dec 2016

**Review of "Seasonal streamflow forecasts for Europe – I. Hindcast verification with pseudo- and real observations" by W. Greuell et al.**

**Reviewed:** December 2016

**Recommendation:** The manuscript is acceptable with minor revisions.

In this paper, the authors present a model-based seasonal hydrological forecasting system, which produces hydrological forecasts for up to seven months of lead time over Europe. As the authors state it, seasonal hydrological forecast systems over Europe are scarce, which makes this work relevant to HESS and to the wider hydro-meteorological community. Furthermore, we are currently at a turning point where model-based dynamical systems are becoming more widely used for seasonal hydrological forecasting. This is because it is only recently that dynamical modelling systems have started becoming at least as skilful as statistical modelling systems or dynamical-statistical hybrid systems. This makes the system presented in this paper state-of-the-art.

The authors analyse the skill of the seasonal runoff and discharge hindcasts against pseudo- and real observations, using a variety of metrics. This complete analysis allows to tackle many aspects of seasonal hydrological forecasting and the results are presented in a pleasant to read and concise way. This paper first demonstrates the levels of predictability reached by this system and the spatiotemporal patterns of skill. From this analysis, the authors have successfully identified regions and periods of high runoff skill. The evaluation also highlights the effect of delay between runoff and discharge on the higher discharge forecasting skill. Furthermore, by doing a comparison between hindcasts verification against pseudo- and real observations (theoretical and actual skill, respectively), the authors have shown that there is a higher theoretical skill than an actual skill in seasonal hydrological forecasting, pointing out the need for actual skill calculations. The last part of the analysis is dedicated to the overview of the metric choice on the results of the analysis, stressing the differences and similarities between the metrics.

The paper is overall clear, written in a generally fluent and precise language, and presents a large quantity of results in a structured and concise way, in a paper of appropriate length for the content. The methods are interesting and give enough details for reproducibility of this work. The paper would nevertheless benefit largely from an improvement of the introduction and the discussion sections, with the aim to set the wider context of this work to the readers.

As a whole, I enjoyed reading this paper and I will therefore be pleased to see it published in HESS. Below are minor comments which will hopefully help the authors to improve the paper.

**Title:** The title is pertinent with regards to the contents of the paper. However, I don't like the terms "pseudo-observations" and "real observations". I would name them differently, such as "analysis" (as done in meteorology) or "simulations", for the pseudo-observations, and simply "observations" for the "real observations".

**Abstract:** Overall, the abstract provides a concise and complete summary of the paper. Here are however a few suggestions that could help clarify certain aspects of the abstract:

- It would be good to say that the hindcasts have 7 months of lead time earlier than on page 1, line 19. This could be mentioned for example in the sentence on page 1, line 15: "Skill is

analysed with a monthly temporal resolution, up to 7 months of lead time, for the entire annual cycle."

- Page 1, line 23: it was not clear to me what the sentence "a conceptual analysis of the two types of verification" meant. Could you please rephrase this to clarify it to the readers here? It could be rephrased to, for example, "attributed to the structural differences between the runs used for the two verification methods."
- Before reading page 1 line 20, it wasn't clear to me that both discharge and runoff were analysed in this paper. It would be good if you could specify it earlier on in the abstract.

**Introduction:** The introduction is interesting, but it could overall contain more literature review on seasonal hydrological forecasting in general: e.g., statistical versus dynamical methods and the state of seasonal hydrological forecasting over Europe, stating the current predictability in Europe (referring to work previously done on the same topic). Here are a few other suggestions that could maybe help to make the introduction more concise.

- Page 1, line 28: the word "may" sounds like society may also not benefit from such forecasts. It would therefore be interesting to refer to papers tackling this topic, such as: Viel et al. (2016), Soares and Dessai (2016), Crochemore et al. (2016), among others.
- Page 1, line 30: it would be good to add references for other applications of the seasonal predictions, as done for the energy generation sector.
- Page 1, line 33: the word "usefulness", just like the word value, is a complex one. Indeed, the usefulness of a system does not only depend on the skill of the forecasts that it produces, but also on the way this skill is transformed into a decision within one of the sectors of interest. This is an interesting post on this topic: https://hepex.irstea.fr/economic-value-of-hydrological-ensemble-forecasts/. I would therefore suggest to change this sentence slightly to acknowledge this complexity in the value of probabilistic forecasts for decision-making, by saying for example: "The usefulness of the system depends partially on […]".
- Page 2, lines 3-4: see my comment for the title of the paper.
- Page 2, lines 6-8: another example of the use of "pseudo-observations" rather than "real observations" is in cases when the aim is to exclude the model error from the analysis in order for example to perform a sensitivity analysis to other components of the forecasting system. For example, the VESPA method introduced in Wood et al. (2016), to look at the contribution of initial hydrological conditions and seasonal climate forecast errors to seasonal streamflow forecast uncertainties. It would be worth mentioning this here.
- Page 2, line 9: you mention that the fact that "pseudo-observations" are not equal to "real observations" is a downside, which is a very good point. This however needs clarification on how it could influence an analysis of the skill of the forecasts here. The sentence on page 2, lines 14-15, could for example be rephrased to sound like a hypothesis and moved earlier.
- Page 2, lines 13-15: this description is already done in the last paragraph of the introduction (page 2, lines 33-34). It is also too methodological for this part of the introduction, which should be more focused on literature review. I would thus suggest to remove it here.
- Page 2, lines 19-23: references to these papers are very interesting. It would be even more interesting if you could also mention results of these analyses briefly, such as answers to the following questions: what is the current predictability in Europe? Where are the high skill areas? Etc.
- Page 2, line 24: could you please add "presented in this paper" after "The hydrological hindcasts"? This would then make it clear what you are talking about.

- Page 2, lines 26-28: could you please state here that the initial hydrological conditions are used for the hindcasts generation?
- Page 2, lines 30-31: could you please specify that this aim is to look at the effects of using "pseudo-observations" for the verification of the hindcasts, as opposed to using "real observations"?
- Page 2, line 34: the sentence about the supplementary figures seems out of place here. I would rather mention in the introduction paragraph of the results section of this paper.
- Page 2, lines 34-40: the results of the companion paper are very interesting but seem out of place here as well. They should either be moved to the discussion section of this paper or mentioned earlier in the introduction, and well linked to the rest of the introduction.

**Section 2.1:**

- Page 3, lines 14-15: what is the time step of these hindcasts? Daily? It would be good to mention it here.
- Page 3, line 15: consider changing the word "simulations" to "forecasts", as it is confusing otherwise.
- Page 3, lines 17-18: could you please specify that these are the System 4 ensembles?
- Page 3, lines 19-24: it would make the lecture of this technical description more structured if this paragraph was combined with the paragraph on page 3, lines 11-13.
- Page 3, line 25: the sentence "and in addition for spin-up periods" could be removed and the following sentence could be linked to the previous to make it clearer. This would then give: "VIC was run for the period of the S4 hindcasts (1981-2010). Additionally, for the reference simulation, two extra years (1979-1980) were run to spin up […]".
- Page 3, lines 30-31: why were the simulations done with a three-hourly time step? It would be good to clarify this here.
- Page 3, lines 37-38: I don't understand what these four other hydrological models are and why they are mentioned here. If they are not used in this paper, I would suggest to remove this piece of the sentence as it might confuse the readers.
- Page 3, lines 39-40: It is interesting to note those aspects as key for seasonal predictions! However, could you please specify what is meant exactly by "more or less in the middle of the ranking of the five models", by for example using scores to support this sentence?

**Section 2.2:**

- Page 4, lines 4-5: how were the data sets converted to gridded versions? It would be useful to mention this here.
- Page 4, line 7: it would be good to mention the area of the grid cells that the catchments cannot pass in order to be considered as "small basins" here.
- Page 4, lines 23-27: what if there are 2 neighbouring cells without an influx from any of the neighbouring cells, corresponding to two small basins? How can we be sure that that nearest cell is in fact that small basin and not the other one?
- Page 4, line 26: this sentence is not entirely clear to me. Do you mean all of the cells with no influx from the eight neighbouring cells?
- Page 4, line 27: is this method appropriate?
- Page 4, lines 29-30: could you please specify that this is over Europe, to remind the reader?

**Section 2.3:**

- Page 4, lines 32-33: it would be good to repeat here again that the analysis was carried out on the 7 months of lead time.
- Page 4, lines 38-39: this explanation is slightly confusing. Could you please rephrase it to make it clearer to the readers?
- Page 5, lines 1-3: from reading the results, forecasts with zero lead time are actually still mentioned a fair amount of times.
- Page 5, line 4: it would be good to specify why you refer the readers to Mason and Stephensen (2008). Is it because they selected the same skill metrics?
- Page 5, line 5: please consider changing the word "simulations" to "forecasts" here.
- Page 5, line 6: what is called the "ROC graph" here is usually called the ROC curve.
- Page 5, lines 7-9: further details are needed for the computation of the ROC score. Please consider providing more details on the following questions: Are the terciles for the ROC computed on the "pseudo-observations"? Are the terciles calculated for each month individually or for the whole period? And from monthly averages? How many bins are used for the ROC?
- Page 5, line 8: the "one third highest, lowest and the remaining values" could simply be called "upper, lower and middle terciles".
- Page 5, lines 9-11: this is vague, it would be nice to talk about attributes of the forecasts and to mention the attributes covered by each metric.
- Page 5, line 11: by "value falling in the considered tercile" do you mean "percentage of ensemble members falling in the considered tercile"?
- Page 5, line 12: it would be good to describe the RPS first, then the RPSS. Also, what is the reference forecast used for the RPSS calculation?
- Page 5, line 13: could you please specify what is meant by "correct forecasts" here? Reliable? Sharp? Accurate?
- Page 5, line 14: is the climatology used as a reference forecast for the measure of skill then?
- Page 5, line 14: by "climatological forecasts (forecasts that are identical each year)", do you mean an ensemble of past historical observations? This is not so clear here.
- Page 5, lines 14-15: could you also please specify what are the best values for each metric. So what value would a perfect forecast have?
- Page 5, lines 19-22: this paragraph should rather be included in the introduction of the results section I think.
- Page 5, line 20: the fact that the correlation coefficient is the easiest to understand is a valid argument. However, it doesn't sound very good to state it here as the primary reason for choosing this metric against others. I would just remove this part of the sentence.
- Page 5, lines 23-24: is it one third of zeros or one sixth of ties over the entire hindcast period? Could you also please justify that rule?

**Section 3:**

- For the results section of this paper, more credit should be given to other papers on seasonal hydrological forecasting in Europe, where appropriate. For example, Crochemore et al. (2016), Demirel et al. (2014), Svensson (2015), Trigo et al. (2004), among others; even if these papers do not contain an analysis for the integrity of Europe.
- Page 5, lines 26-30: this description was already made in the introduction. I wouldn't repeat it here, especially since the results section titles are quite descriptive.

**Section 3.1:**

- Page 5, lines 39-40: this is a very interesting remark!
- page 5, lines 32-40: how are those results different or similar to results for the other initialisation months?
- Page 6, lines 18-19: this figure does however not look at the persistence in skill, as a single cell could have skill for 3 months in a row for example, and another for 3 months but spaced, having the same colour on figure 3. It would be worth mentioning this in the figure caption.
- Page 6, lines 27-28: that is a very interesting results. Would it be possible to say why this is? Are cells in a specific region gaining skill or is it random noise?
- Page 6, lines 28-30: a result worth mentioning however, would be the lead time at which, on average, the domain-averaged R ≤ 0.

**Section 3.2:**

- Page 6, line 34: could you please add "(not shown)" at the end of the sentence finishing with "target months and lead times."?
- Page 7, line 8: could you please add the word "difference" after "average"?
- Page 7, lines 15-16: this is a good point!

**Section 3.3:**

- Page 7, lines 23-24: was the same observed for other initialisation and target months? It would be good to mention this here.
- Page 7, lines 23-26: with this sample of stations, is it possible to say if there are regions where the difference between theoretical and actual skill is highest?
- Page 7, lines 32-40: this paragraph describes methods and should therefore be moved to the methods of analysis section of this paper.
- Page 8, lines 1-3: what about basins with an AAPFD > 0.3? they would probably show a higher difference between the two ratios.
- Page 8, line 10: could you please add the word "observation" before "stations"?
- Page 8, line 13: is the skill reduction between theoretical and actual skill or between lead time 0 and 2? The following sentence suggests that it is the former but it is not clear from the sentence so it would be good to specify.
- Page 8, lines 13-17: this is very interesting!

**Section 3.4:**

- Page 8, line 20: it would be good to specify that we are looking at runoff again here.
- Page 8, lines 20-21: did the other initialisation and target months show similar results? It would be good to mention this here.
- Page 8, line 23: I am not sure to understand the sentence "domain-averaged magnitude of the skill metrics". Could you please clarify what is meant?
- Page 8, lines 22-24: the patterns of skill are indeed similar. However, the magnitudes appear fairly different, even given the fact that they cannot be compared exactly due to the different colour bars used for plotting. The RPSS for example shows a lower skill on average than the other scores, while R shows a higher skill on average. This is also shown by the cells signal for each score. It would be worth noting this, and also in terms of the forecast attributes.
- Page 8, line 29: yes, this can be done with the cell signal indicated on the top left corners of the plots.

- Page 8, lines 30-32: this is very interesting. So it indeed suggests that seasonal forecasts are anomaly forecasts, which is useful for decision-making! How are those numbers equal or different for other target and initialisation months?
- Page 8, line 35: I would rephrase the explanation of the RPSS here.
- Page 8, line 38: which results is this referring to? All the results presented in this section so far? Could you please specify it here or mention it little by little after each result?
- Page 8, lines 39-40: this is not true for all cases, but it is on average.
- Page 8, line 41: is the 1.0 here in terms of the ROC area?

**Discussion:** the differences between the theoretical and the actual skill stated here are very interesting. However, the discussion would benefit greatly from further examples on how to improve the actual skill of seasonal hydrological forecasts (such as the recalibration idea given on page 10, lines 29-33).

- Page 9, lines 4-12: this should be moved to the methods, together with Figure 10. Then in the discussion you could refer back to these structural differences between the systems and state the questions that these differences raise.
- Page 9, line 34: could you please rephrase the sentence "In the real world a difference discharge observations differ from reality"? It is not clear to me what is meant.
- Page 9: lines 34-36: this is an interesting point. However, I do not see how it will lead to more theoretical than actual skill. Indeed, the bias in the discharge measurements could potentially mean a closer simulated discharge from the model reference run to the biased discharge observations. In other words, we do not know how this measurement bias impacts the actual skill with regards to the theoretical skill.
- Page 9, lines 37-42: it would be clearer if you made this point number 4, even if this component on Figure 10 is not in red.
- Page 10, lines 4-12: this is a very good point! I would put it in the model hydrology box, so within point 2 on page 9, or as a sub-point of point 2.
- Page 10, line 13: could you please add (see Sect. 3.3) in parentheses at the end of this sentence?
- Page 10, line 17-18: I don't understand why this would be the case? The hindcasts would also benefit from the model optimisation as they are run with the same model as the reference run. The only difference between those two systems being the meteorological forcing data used to produce the hindcasts or pseudo-observations of discharge.
- Page 10, lines 14-28: I am not sure to understand the point that you are making here. The model is the same for the reference run and the hindcasts generation, hence, even if the model is optimised to reach closer discharge simulations to the actual discharge observations, both systems would benefit from this. In the examples that you give, the predictive skill gained from wrongly forecasting this too large amount of snow or soil moisture runoff or from rightfully forecasting lower snow or soil moisture runoff should be the same, unless the metric used to calculate skill is biased towards large values, such as the MAE, for example. So the problem here is rather the choice of the metric. In case I am missing something, could you please clarify this paragraph?

**Conclusions:**

- Page 11, lines 9-10: please consider adding a "for example" here to show that the British Isles are an example amongst many results of the paper.
- Page 11, lines 10-11: is this true for all lead times?

- Page 11, line 19: I wouldn't mention the numbers in between parentheses in the conclusion. They are already in the results of the paper, where the readers can find them if they want to.
- Page 11, lines 21-22: I would write the Ranked Probability Skill Score as RPSS since ROC is also written as an abbreviation.
- Page 11, lines 22-23: could you please rephrase this sentence to "The skill in terms of the ROC area tends to be slightly larger for […]"?

**Figure 1:** these are great plots!

- Could you please put a label on the side of the colour bar to indicate that this is R?
- Please state in the figure caption that red is better.
- Could you please specify that the legend is situated in the top left corner of each plot? This is a really good idea by the way!

**Figure 2:**

- Could you please put a label on the side of the colour bar to indicate that this is R?
- Even though the caption is given in Figure 1, I would repeat it here. Because it is easier to read directly under the figure than having to jump from a figure caption to the other figure.

**Figure 3:**

- Could you please put a label on the side of the colour bar to indicate that this is R?

**Figure 4:** this figure is great, I especially like the lead time lines, clever!

- Could this figure be made bigger?
- In order to make it easier to read for the readers, please consider adding a colour bar for the different initialisation months.

**Figure 5:**

- I would put the a, b, c and d above each plot.
- Wouldn't it be better and easier to see the differences between plots a and b if a plot of the difference between both maps was made instead?
- In the y-axis labels of plots c and d the word correlation coefficient can be replaced with R.
- Could you please add an x-axis label for plot c to say if these are the initialisation or target months?
- Plot d is not colour blind friendly as there is both red and green. Please consider changing one of the two colours.
- Here again I would repeat the necessary information of the caption of Figure 1 for the interpretation of this figure.
- It would be good to specify the amount of catchments in each bin for plot d. This could maybe explain the negative difference for bin 8 for lead times 2 and 4.
- Could you please put a label on the side of the colour bar for plots a and b, to indicate that this is R?

**Figure 6:**

- I would put the a, b, c, d and e above each plot.
- Why isn't there a plot for the "real observations" and all stations for May and lead time 2? It would be interesting to see I think.

- Could you please put a label on the side of the colour bar to indicate that this is R?
- Could you remind the readers what the sizes of small and large basins are in the caption, as well as the number of stations for both categories?

**Figure 7:**

- Could you please put letters for both plots here: a and b?
- In the y-axis labels of both plots the word correlation coefficient can be replaced with R.
- I would remove the y-axis label and the tick labels of the second figure as it is already stated in the figure on the left.
- Could you please add an x-axis label for both plots to say if these are the initialisation or target months?
- Could you please remind the readers what the sizes of small and large basins are in the caption, as well as the number of stations for both categories?

**Figure 8:**

- I would put the a, b, c and d above each plot.
- Because plot d does not show much and this paper already contains many figures, would it be maybe better to remove plot d and mention it in the text only? Figure 9 could then replace plot d for example.

**Figure 9:**

- Could you please add an x-axis label for both plots to say if these are the initialisation or target months?
- Would it be worth adding lines for the middle tercile in this same plot?
- Page 17, line 10: I think that the "minus" does not belong here.

**Technical corrections:**

- General:
    - Could you please only use of the two terms: "basins" or "catchments"?
    - Please consider changing "lead month" to "lead time", which is more widely used, and will hence be clearer for the readers even without having read the methods section.
    - Could you please replace "panel" with Fig. figure# subfigure#? E.g., for Figure 5, panel c would be replaced by Fig. 5c.
    - Could you please consider renaming the terms "pseudo-observations" and "real observations"? I would for example use "analysis" (as done in meteorology) or "simulations", for the pseudo-observations, and simply "observations" for the "real observations".
    - Could you please change "North" to "Northern", "South" to "Southern", "West" to "Western" and "East" to "Eastern" when in front of a country's name?
- Page 1, line 10; page 11, line 3: please consider rephrasing the sentence "The present paper presents […]" by removing one of the words "present".
- Page 1, line 26: the terms "below normal" and "above normal" should not be written with capital letters, unless the abbreviations "BN" and "AN" are given in between parentheses just after.
- Page 1, line 29; page 2, lines 2 and 7; page 3, line 7: "e.g." should be replaced with ", for example,".

- Page 2, line 11: either "like" or "e.g." should be used here.
- Page 2, line 12: please consider changing the word "earlier" by for example "previously".
- Page 3, lines 8 and 9: please consider changing one of the two words "namely" to a synonym of this word.
- Page 3, line 10: "which is then used for" the "initialisation of the hindcasts".
- Page 3, line 12: please remove the word "again".
- Page 3, line 12: does "here" mean "hereafter as"?
- Page 3, lines 16-17: this should be moved to the references section of this paper and cited here.
- Page 3, line 30: please change "Though" to "Although".
- Page 4, line 6: should the hyphen be removed between the words "large" and "basins"?
- Page 6, lines 34-35: please rephrase this sentence to "There are however subtle differences because rivers […]".
- Page 7, line 9: "the rate with which" instead of "the rate by which".
- Page 7, line 38: a ";" should be added between "AAPFD" and "see Marchant and Hehir, 2002".
- Page 7, line 39: this should be "AAPFD". The D is missing.
- Page 7, line 42: the "So" is not needed here.
- Page 8, line 6: there should be a "," between "R" and "theoretical" to clarify the sentence.
- Page 8, line 10: "can be blamed on" rather than "to".
- Page 8, line 14: there is a "to" missing between "due" and "a combination".
- Page 9, lines 24, 36 and 42: please remove the "to" between the words "than" and "actual".
- Page 9, line 29: please put the "see the companion paper" between parentheses.
- Page 10, line 5: please put the "see Sect 2.2" between parentheses.
- Page 10, lines 5-6: please consider changing the second "differences" in the sentence to for example "disagreement".
- Page 11, lines 3-4: please consider adding the word "while" between just after the comma, to link the two parts of the sentence.
- Page 11, line 5: would replacing "taking" with "against" make more sense here?
- Page 11, line 5: please consider replacing the "as" with "called".

---

## Referee Comment (RC2) · C. Prudhomme (Referee) · 9 Feb 2017

General

The paper is the first of 2 companion papers on a pan-European seasonal streamflow forecasting system. This paper focuses on the verification of the re-forecast for a 30-year period (1981-2010).

Streamflow forecasting beyond medium range is still a relatively new area of research in Europe, and has received more attention in the past few years, following the availability of seasonal climate re-forecasts. Skilful hydrological forecasts at monthly to seasonal lead time would have great potential use in Europe as it would help planning

and management of water resources for a huge variety of sectors including transportation, agriculture, public and domestic water supply or energy. Whilst the skill of dynamic rainfall forecasts is relatively limited at lead times over 10 days in temperate climates such as Europe, the existence hydrological memory due to catchment storage raised the question of potential higher skill in hydrological seasonal forecast than in its climate forcing data. As such, the paper addresses a topical subject with a large readership interest. I have however some concerns about some of the analyses undertaken here, detailed below. I hence suggest a major revision.

The streamflow forecasting system developed and used in this paper relies on two major sources of information and tool: 1) climate forcing data, here based on the ECMWF System 4 re-forecasts; and 2) a gridded hydrological model that transforms the weather signal into runoff and routed discharge. Inherent to any modelling exercise, simulations and re-forecasts are likely to be associated with bias and errors.

The authors run a gridded hydrological model forced by observed climate for 1 month, as spin-up to set-up initial conditions, and run the model with re-forecast climate forcing. They then evaluate the skill of the re-forecasts by comparing the results with 1) hydrological simulations forced by observed climate (runoff and routed discharge; called 'theoretical skill'); 2) observed discharge (called 'actual skill'). For actual skill, they use discharge time series from the GRDC and EWA database, and match the location of the river gauges with the routed network used in the model (at a 0.5°x0.5°resolution, i.e. ~ 50km) so that gauged flows can be compared with the correct modelled discharge. Three metrics are used for the theoretical skill assessment, but most discussion is based on correlation coefficients, also applied to actual skill. The seasonal variation of the spatial distribution of the theoretical skill is described and compared for runoff and discharge, mainly for a 2-month lead time. Overall pan-European theoretical and actual skill compared for 2 classes of catchment size, and some causes of degradation between theoretical and actual skill discussed, but not formally tested.

Whilst the findings of pan-european hydrological seasonal forecasting skill are really

relevant, I have some reservation regarding some methodological decisions and interpretations presented in the paper, detailed below.

- Actual skill analysis. The analysis must be better justified, and the discussion strengthened. Below are some points that need to be added to the paper:

o Is simulated discharge comparable to actual discharge? There is no data assimilation at the beginning of the forecast to reduce potential bias in the simulated discharge. So the hydrological re-forecasts include both hydrological modelling errors and climate forcing errors, without any attempt to reduce the former.

o Is the catchment matching exercise working? The hydrological model has a relatively coarse resolution, and a catchment area error of up to 15% (for large catchments) is deemed acceptable [the choice of this threshold should be justified]. For small catchments, there is no attempt to scale the discharge from the hydrological model scale to the gauged catchment scale. This could introduce some discrepancies between simulated and observed discharge. In fact p8 l3-4, the authors do state that '[the] small basins (. . .) are generally smaller than the spatial resolution of the simulations'

o Is the hydrological model performance influencing the actual skill results? Poor hydrological model performance will introduce errors for both initial states and re-forecasts. One hypothesis is for 'actual skill' to be much lower for seasons and locations where the hydrological model is known not to reproduce well the hydrological processes. Comparison of hydrological model performance and actual skill is necessary for a meaningful interpretation of the results. This is only mentioned briefly in the discussion (2.5 lines) as second point (p9 l31-33). This should be the first point of the analysis when regarding actual skill.

- Re-forecast simulations

o Is the spin-up period long enough? It is not clear what actual spin up is used, with 1-month spin-up period suggested (p3 l29), but this sounds really short compared to

expected storage in some parts of Europe (e.g. snow pack in high latitude/ high elevation and/or groundwater storage in large aquifers).

- General methodology

o How are the catchments classified as small/ large? There is no surface area mentioned, and not physical justification, but size is the only physical measure used to attempt explaining the difference between theoretical and actual skill

o What is the justification for the non-calculation of skill metrics? (p5l23-24). In particular, zero flow simulations can be extremely important to depict droughts. Why excluding them?

o How is a skilful forecast defined? (p5 l37-38; p6 l1-2) What is the threshold used to define a re-forecast as 'skilful'? Is this based on statistically significance test? Is it the value of 0.31 quoted in caption fig 1? This needs to be made clearer within the text

o Human influence analysis. This is fully based on the assumption that LPJmL has identified and reproduces accurately all the human interventions, and the derived Amended Annual Proportional Flow Deviator is a realistic representation of the degree of influence. This is a strong assumption that needs to be caveated in the text. This modelling exercise needs to be described in the methods section and not so late in the paper (p7 l34-36)

- Analysis/ interpretation

o Influence of catchment size on theoretical vs actual skill (p8 l4-17). I found the analysis difficult to follow, the paragraph confusing, and the language used is inappropriate 'apparent difference in (. . .) skill (. . .) can be blamed almost entirely to the geographical distribution of stations'. What does 'this results holds for the cells with observations' mean? Is the difference between 'large basins' skills (0.396) and 'small basins' skills (0.384) significant? Is this to be linked with the scale of the hydrological modelling? The analysis would be more thorough if conducted by looking at relationships with

catchment sizes, rather than dividing the sample in 2 categories. It also needs to be linked with the modelling performance.

o Section 3.4 (p8). Is this conducted on pseudo-observations? Why is this not after section 3.2? What is the implication of the findings? Can a physical explanation be given? Can the authors recommend skill metrics following their analysis?

o Discussion (p9-10). I found it unclear and difficult to follow, and some description of methods (model calibration technique) don't fit well (this should be in methods). The authors here describe some hypotheses for the difference between theoretical and actual skill: this should come at the beginning of the paper, and being tested within the study. Moreover, the analysis between theoretical and actual skill is short and not very thorough, yet is discussed at length; this does not reflect well the study. Some points are not clear (e.g. p9 l26-30; p9 l39-42)

o Statements not justified. There is a lack of evidence of the authors' claim that 'optimisation of the model system could, and would in many case, lead to a degradation of the theoretical skill'. What is the reason for that? What is the evidence? Have the authors conducted a sensitivity analysis? I agree that perfect theoretical skill does not adequate with perfect re-forecast, when main processes are not accounted for in the models. But the whole section needs careful re-wording, and better scientific justification, references, or suggestions for further analysis for verification of the hypotheses.

Main points of suggested improvement

Science

- There is no information on the hydrological model performance, albeit it is written to be 'on average across all basins considered, more or less in the middle ranking of the five models' [p3l39-40]. This is not enough and does not provide any information of the actual performance (it could be middle ranking of an ensemble with very low skill). Reference of a paper is not enough in this case. This is critically important when

the re-forecast skills are compared with what the authors call real- observations, as it would be expected that lower hydrological modelling performance would result in lower skill in reproducing the real observations.

- There is not enough discussion on the role of initial conditions, hydrological memory and catchment storage that can bring predictability: catchment storage could include groundwater, lakes, and snow pack. At the very least, reference to some of the findings of part 2 could be made.

- There is a lot of discussion about the quality of measurements and their implication on lower actual skill, and much less on modelling error. I found this out of proportion.

- Current conclusion is a summary of the research. I would expect the discussion to be opened to future research and/or application.

- The reference to the companion paper (page 2) is very limited, and it is difficult to see the link between both. At least the conclusions could be brought in the discussion, rather than exposed in the introduction and not referred to later onto justify the writing up of the study in 2 parts.

Structure

- The title does reflect the bulk of the paper. The analysis of 'real-discharge' is only done in section 3.3, out of 4 analysis sections.

- The structure is not logic: 3.1, 3.2 and 3.4 all analyse the results in a 'pseudo-observations' [modelled] world, whilst 3.3 looks at the results in 'real- observations' world.

- Description of the model set-up/ calibration is given in the discussion (p10 l29-33), but this should be in the methods section when the model is introduced

Other points

Science

[Figure]

- The explanation of matching gauges locations with the 0.5 grid needs to be improved

Structure/ description

- Introduction. Most of the introduction is dedicated to the methods, data and tools used in the paper, and is not a review and discussion of the state of the art, with a judgment of the conclusions obtained from previous studies, and how to move forward. A typical example is p2 l9-15, with a list of papers without any discussion, and a description of some of the analysis, and even a discussion of the results, which should not be in introduction. I found this very confusing. The whole section needs to be greatly improved, with a more traditional layout of state of the art, research gaps identified, and then at the end aims of the paper, without details of the methods and tools used.

- Section 3.1: Inconsistency in figure references; first sentence of page 6 does not describe what fig 2 shows.

- Figure 3 is excellent.

---

## Author Comment (AC2) · 9 Mar 2017

Referee #2 Christel Prudhomme (hess-2016-603-RC2)

We thank Christel for her more critical (than RC1) but very constructive review. Some of her remarks are in line with those of RC1, some are additional. Below we will discuss the main remarks, details can be found in hess-2016-603-RC2-author-reply.pdf.

A number of remarks have to do with the structure of the paper: - RC2 requests a better description of the main findings of cited literature mostly in the introduction, but also in other parts of the paper, and how these influenced the objectives of our study. We now recognize that this indeed can, should and will be improved, together with some

additional references as suggested by RC1. - With RC1, RC2 suggests to move the first part of the discussion, the part explaining the present figure 10 to the Methodology section. We will do so. - RC2 asks repeatedly for suggestions/recommendations for further analysis. We recognize this omission, but of course have thought about that rather extensively. We will add such suggestions where appropriate in the discussion As a result of these 3 points, and some others, both the introduction and the discussion section will be largely rewritten.

Some remarks pertain more to the science of our analysis: - RC2 asks for a better description of the deterministic performance of the model used (VIC) prior to its use in a probabilistic seasonal forecasting context. We will do so based on and referring to previously published work, both from our own group (Greuell et al. 2015, Haddeland et al., 2012; van Vliet et al. 2012) and from others. More in particular we will try to relate good and bad forecasting skill for certain regions/basins and seasons in Europe to previously identified strengths and weaknesses in VIC performance , i.e. strengths/weaknesses to reproduce historical river flows across Europe. - This issue partially overlaps with the RC2 request to better analyse the potential relation between basin size and model hindcast skill. Without focussing on individual basins (which is one the directions for future work we'd like to take). We will prepare, present and discuss a graph similar to the present Fig 5d, but then relating difference between actual and theoretical discharge skill to basin size. This will be a new piece of analysis leading to a yet unknown outcome. Thus we will also increase the relative importance of section 3.3 better justifying the title of this paper.

Altogether, we believe that by following most of the recommendations by both RC1 and RC2 we will be able to significantly improve the structure and readability of the paper, as well as improving the scientific quality by some additional analysis and especially much better 'embedding' in previous work, both our own and that of others. Finally, priori to resubmission we'll have a language check done by a native speaker.

**HESSD**

Please also note the supplement to this comment:
http://www.hydrol-earth-syst-sci-discuss.net/hess-2016-603/hess-2016-603-AC2-supplement.pdf

─────────────────────────────

Interactive
comment

**Supplement:**

General

The paper is the first of 2 companion papers on a pan-European seasonal streamflow forecasting system. This paper focuses on the verification of the re-forecast for a 30-year period (1981-2010).

Streamflow forecasting beyond medium range is still a relatively new area of research in Europe, and has received more attention in the past few years, following the availability of seasonal climate re-forecasts. Skilful hydrological forecasts at monthly to seasonal lead time would have great potential use in Europe as it would help planning

and management of water resources for a huge variety of sectors including transportation, agriculture, public and domestic water supply or energy. Whilst the skill of dynamic rainfall forecasts is relatively limited at lead times over 10 days in temperate climates such as Europe, the existence hydrological memory due to catchment storage raised the question of potential higher skill in hydrological seasonal forecast than in its climate forcing data. As such, the paper addresses a topical subject with a large readership interest. I have however some concerns about some of the analyses undertaken here, detailed below. I hence suggest a major revision.

The streamflow forecasting system developed and used in this paper relies on two major sources of information and tool: 1) climate forcing data, here based on the ECMWF System 4 re-forecasts; and 2) a gridded hydrological model that transforms the weather signal into runoff and routed discharge. Inherent to any modelling exercise, simulations and re-forecasts are likely to be associated with bias and errors.

The authors run a gridded hydrological model forced by observed climate for 1 month, as spin-up to set-up initial conditions, and run the model with re-forecast climate forcing. They then evaluate the skill of the re-forecasts by comparing the results with 1) hydrological simulations forced by observed climate (runoff and routed discharge; called 'theoretical skill'); 2) observed discharge (called 'actual skill'). For actual skill, they use discharge time series from the GRDC and EWA database, and match the location of the river gauges with the routed network used in the model (at a 0.5°x0.5° resolution, i.e. ~ 50km) so that gauged flows can be compared with the correct modelled discharge. Three metrics are used for the theoretical skill assessment, but most discussion is based on correlation coefficients, also applied to actual skill. The seasonal variation of the spatial distribution of the theoretical skill is described and compared for runoff and discharge, mainly for a 2-month lead time. Overall pan-European theoretical and actual skill compared for 2 classes of catchment size, and some causes of degradation between theoretical and actual skill discussed, but not formally tested.

Whilst the findings of pan-european hydrological seasonal forecasting skill are really

relevant, I have some reservation regarding some methodological decisions and interpretations presented in the paper, detailed below.

- Actual skill analysis. The analysis must be better justified, and the discussion strengthened. Below are some points that need to be added to the paper:

o Is simulated discharge comparable to actual discharge? There is no data assimilation at the beginning of the forecast to reduce potential bias in the simulated discharge. So the hydrological re-forecasts include both hydrological modelling errors and climate forcing errors, without any attempt to reduce the former.

o Is the catchment matching exercise working? The hydrological model has a relatively coarse resolution, and a catchment area error of up to 15% (for large catchments) is deemed acceptable [the choice of this threshold should be justified]. For small catchments, there is no attempt to scale the discharge from the hydrological model scale to the gauged catchment scale. This could introduce some discrepancies between simulated and observed discharge. In fact p8 l3-4, the authors do state that '[the] small basins (…) are generally smaller than the spatial resolution of the simulations'

o Is the hydrological model performance influencing the actual skill results? Poor hydrological model performance will introduce errors for both initial states and re-forecasts. One hypothesis is for 'actual skill' to be much lower for seasons and locations where the hydrological model is known not to reproduce well the hydrological processes. Comparison of hydrological model performance and actual skill is necessary for a meaningful interpretation of the results. This is only mentioned briefly in the discussion (2.5 lines) as second point (p9 l31-33). This should be the first point of the analysis when regarding actual skill.

- Re-forecast simulations

o Is the spin-up period long enough? It is not clear what actual spin up is used, with 1-month spin-up period suggested (p3 l29), but this sounds really short compared to

expected storage in some parts of Europe (e.g. snow pack in high latitude/ high elevation and/or groundwater storage in large aquifers).

- General methodology

o How are the catchments classified as small/ large? There is no surface area mentioned, and not physical justification, but size is the only physical measure used to attempt explaining the difference between theoretical and actual skill

o What is the justification for the non-calculation of skill metrics? (p5l23-24). In particular, zero flow simulations can be extremely important to depict droughts. Why excluding them?

o How is a skilful forecast defined? (p5 l37-38; p6 l1-2) What is the threshold used to define a re-forecast as 'skilful'? Is this based on statistically significance test? Is it the value of 0.31 quoted in caption fig 1? This needs to be made clearer within the text

o Human influence analysis. This is fully based on the assumption that LPJmL has identified and reproduces accurately all the human interventions, and the derived Amended Annual Proportional Flow Deviator is a realistic representation of the degree of influence. This is a strong assumption that needs to be caveated in the text. This modelling exercise needs to be described in the methods section and not so late in the paper (p7 l34-36)

- Analysis/ interpretation

o Influence of catchment size on theoretical vs actual skill (p8 l4-17). I found the analysis difficult to follow, the paragraph confusing, and the language used is inappropriate 'apparent difference in (…) skill (…) can be blamed almost entirely to the geographical distribution of stations'. What does 'this results holds for the cells with observations' mean? Is the difference between 'large basins' skills (0.396) and 'small basins' skills (0.384) significant? Is this to be linked with the scale of the hydrological modelling? The analysis would be more thorough if conducted by looking at relationships with

catchment sizes, rather than dividing the sample in 2 categories. It also needs to be linked with the modeling performance.

o Section 3.4 (p8). Is this conducted on pseudo observations? Why is this not after section 3.2? What is the implication of the findings? Can a physical explanation be given? Can the authors recommend skill metrics following their analysis?

o Discussion (p9-10). I found it unclear and difficult to follow, and some description of methods (model calibration technique) don't fit well (this should be in methods). The authors here describe some hypotheses for the difference between theoretical and actual skill: this should come at the beginning of the paper, and being tested within the study. Moreover, the analysis between theoretical and actual skill is short and not very thorough, yet is discussed at length; this does not reflect well the study. Some points are not clear (e.g. p9 l26-30; p9 l39-42)

o Statements not justified. There is a lack of evidence of the authors' claim that 'optimisation of the model system could, and would in many case, lead to a degradation of the theoretical skill'. What is the reason for that? What is the evidence? Have the authors conducted a sensitivity analysis? I agree that perfect theoretical skill does not adequate with perfect re-forecast, when main processes are not accounted for in the models. But the whole section needs careful re-wording, and better scientific justification, references, or suggestions for further analysis for verification of the hypotheses.

Main points of suggested improvement

Science

- There is no information on the hydrological model performance, albeit it is written to be 'on average across all basins considered, more or less in the middle ranking of the five models' [p3l39-40]. This is not enough and does not provide any information of the actual performance (it could be middle ranking of an ensemble with very low skill). Reference of a paper is not enough in this case. This is critically important when

the re-forecast skills are compared with what the authors call real- observations, as it would be expected that lower hydrological modelling performance would result in lower skill in reproducing the real observations.

- There is not enough discussion on the role of initial conditions, hydrological memory and catchment storage that can bring predictability: catchment storage could include groundwater, lakes, and snow pack. At the very least, reference to some of the findings of part 2 could be made.

- There is a lot of discussion about the quality of measurements and their implication on lower actual skill, and much less on modelling error. I found this out of proportion.

- Current conclusion is a summary of the research. I would expect the discussion to be opened to future research and to application.

- The reference to the companion paper (page 2) is very limited, and it is difficult to see the link between both. At least the conclusions could be brought in the discussion, rather than exposed in the introduction and not referred to later onto justify the writing up of the study in 2 parts.

Structure

- The title does reflect the bulk of the paper. The analysis of 'real-discharge' is only done in section 3.3, but out of 4 analysis sections.

- The structure is not logic: 3.1, 3.2 and 3.4 all analyse the results in a 'pseudo-observations' [modelled] world whilst 3.3 looks at the results in 'real- observations' world.

- Description of the model set-up/ calibration is given in the discussion (p10 l29-33), but this should be in the methods section when the model is introduced

Other points

Science

- The explanation of matching gauges locations with the 0.5 grid needs to be improved

Structure/ description

- Introduction. Most of the introduction is dedicated to the methods, data and tools used in the paper, and is not a review and discussion of the state of the art, with a judgment of the conclusions obtained from previous studies, and how to move forward. A typical example is p2 l9-15, with a list of papers without any discussion, and a description of some of the analysis, and even a discussion of the results, which should not be in introduction. I found this very confusing. The whole section needs to be greatly improved, with a more traditional layout of state of the art, research gaps identified, and then at the end aims of the paper, without details of the methods and tools used.

- Section 3.1: Inconsistency in figure references; first sentence of page 6 does not describe what fig 2 shows.

- Figure 3 is excellent.

---

## Referee Report (RR1)

**Review of "Seasonal streamflow forecasts for Europe - I. Hindcast verification with pseudo- and real observations"**

By Wouter Greuell et al.

**Summary:**

This is a detailed analysis of the quality of the pan-European WUSHP seasonal runoff and discharge forecasts, measured against pseudo- and real observations. The analysis is complete and elaborate and supports the authors' conclusions. The paper could however benefit from being more logically structured, as well as more concise in places. I therefore recommend this paper to be published after minor revisions from the authors.

**Major comments:**

1) References to other studies should be concise in the introduction, methods and results sections of the paper. Results from the relevant literature should be described and discussed more fully in the discussion, in order to mention how these results compare to the results presented in this paper. For example: on P3L80-P4L116, P8L287-P9L315 and P16L530-550.

2) This paper contains plenty of interesting and significant results which are overall clearly described. However, the varying choice of target month and lead month used as examples in the results appears unjustified (e.g. April initialisation for Fig.2 vs. May initialisation for Fig. 6a and 6b, lead month 2 for Fig. 3, 4, 6a and 6b vs lead months 0 to 2 for Fig. 5 and lead months 0 to 4 for Fig. 6c, etc). Why aren't the same months display all the time? Could you please justify your choice here and/or in the paper?

3) The (dis)similarity of the results for other initialisation months/lead months/variables is not always discussed, and the relevant supplementary figures are not always referred to in the text. For example on P12L457-467: how do the results from Fig. 2 compare to other initialisation months? No reference is made to the corresponding supplementary figures. On P14L484-512: how do the results from Fig. 3 compare to other lead months? No reference is made to the corresponding supplementary figures. On P15L519-P16L528: how do the results from Fig. 4 compare to other lead months? I don't see any supplementary figures for this figure, why is that? In Section 3.4, the results for other skill metrics are shown and discussed for runoff only. How do these compare to the results for discharge? Can similar conclusions be made?

4) I find the presentation of some of the results not very straightforward. Indeed, in some instances, an example is used to describe the results in details, before giving a general overview of the results. For example, I would have found it more logical to discuss and show the general overview of skill through time with Fig. 5 before presenting the spatiotemporal variation in skill with selected examples (specific initialisation and lead months) presented in Fig. 2, 3 and 4. On P18L600-P19L614: I would have found it here once again more logical to discuss and show the general overview of the difference between discharge and runoff skill (Fig. 6c and d) before discussing the specific examples contained in Fig. 6a and b.

**Specific (minor) comments:**

**-** The abstract is currently too detailed and would greatly benefit from being shortened.

- It is confusing to read the word simulation referring to the hindcast runs. E.g. P2L22 ("hydrology is simulated") or P2L25 ("hindcast simulations"), and throughout the paper. Please rephrase those instances.

- P2L27: you mention discharge but not runoff, which should probably also be mentioned here.

- P2L43-44: the sentence "which is consistent […] two types of verification." does not make sense to me. Consider removing it as not crucial here or rephrasing it.

- P3L62: I am not sure what is meant by "legitimate" here, consider choosing another word.

- P3L63-70: I would move the few lines "In this paper […] adopted here as well" to the end of the introduction, before you present the aims of the paper, and after you have introduced the subject area.

- P3L76: consider adding the paper by Arnal et al. (in review), under review in this special issue, to your list of references for European studies.

- P3L80-81: are the meteorological hindcasts used by Thober et al. (2015) seasonal? Please specify.

- P3L93-P4L102: the skill of which variable(s) is this referring to?

- P4L100: you haven't introduced what NAO stands for.

- P4L120: please specify what the timescale of "medium range forecasting" is.

- P4L128-130: using the words "misleadingly" and "more appropriately" is not appropriate here. Please remove them and simply mention the different names given to pseudo-observations by using more objective words, such as "also called …"

- P4L140: the assessment of skill from pseudo-observations does not exclude model errors per se. Indeed, model errors are variable depending on the magnitude of the flow and can hence be different between the pseudo-observations and the hindcasts. Please consider rephrasing this sentence to reflect this.

- P5L142: please add the reference to the companion paper here.

- P5L145-154: I would suggest to move the sentences that relate to this paper's methods from this paragraph to the following paragraph. The hypothesis could even be moved to the methods section of this paper. The reference to literature where both theoretical and actual skill are assessed can however be kept here.

- P5L158: replace "maximum resolution, i.e. at monthly resolution" with "a monthly resolution" simply. It is confusing what maximum means and the resolution could be higher than monthly aggregations.

- P5L160: replace "for the more" with "using". Using the term "more" does not make sense here, as the correlation coefficient is not a probabilistic metric in the first place.

- P5L173-179: consider moving this paragraph to a more suitable place, e.g. Section 2.3.

- P6L211-212: the point of this sentence is not clear. I would replace it with the hypothesis from P5L151-154, which makes a clear point on the impact that this may have on the results.

- P7L245: please specify which parameters of the WFDEI are used.

- P8L269: it would be good to cite one or multiple papers that support the statement that bias correction generally improves forecasting skill.

- P8L277-279: these lines can be removed here as already previously said in this section.

- P9L324: I suppose "size" refers to the basin size?

- P10L372: I would be interested to know more about the AAPFD. Could you please provide one or two lines of explanation as to what this is?

- P11L392: it might not be clear to some readers what target and lead months refer to. Adding a short explanation of those terms here would be good.

- P11L398-399: please note that this explanation does not work >12, consider rephrasing.

- P11L401: reiterate what the benchmark for the measurement of skill is here.

- P11L405-409: it is not clear to me whether the observation terciles or the hindcast terciles were chosen as thresholds to calculate the forecast probabilities used in the ROC area computation.

- P11L422: I believe the RPSSS is not only a measure of forecast accuracy. It is also for example sensitive to the forecast sharpness.

- P12L443-445: why are hindcasts or observations with more than $1/6^{th}$ of ties discarded?

- P12L465-467: although there is indeed a general impression that skill is fading, it does not appear entirely true for parts of Fennoscandia. What could it be due to?

- P14L487-512: these points would be easier to read if summarised in a table.

- P15L500 and P15L509: "up to lead month 2" does not make entire sense when talking about results from Fig. 3, as it only displays results for lead month 2. Please rephrase or refer to supplementary figures.

- P19L612: mention where the Loire is, as not every reader will be familiar with this river.

- P20L666-667: the reduction in actual skill for Fennoscandia also appears true for small river basins.

- P20L670-P21L671: in Central Europe, some useful skill remains, both for small and large basins, when using real observations.

- From Fig. 7d and e, it seems that there is more skill when using real observations than pseudo-observations for small basins in NW France and SE England. What could that be due to?

- P21L675-P22L712: does the difference between theoretical and actual skill keep on decreasing for lead months > 2?

- P21L677: please specify what the reduction in skill refers to: i.e. between theoretical and actual skill.

- P21L680-681: the exact numbers which produce the ratios are not necessary to mention.

- P22L704-705: this appears evident for lead month 0 but not for lead month 2.

- P22L709-710: this sentence is not completely clear. Consider rephrasing to for example: "there is a linear relationship between the ratio of actual to theoretical skill and the hindcast skill.

- P22L745: I would remove this part of the sentence ("and thus such cases are likely due to chance"). The skill metrics all capture different attributes of the forecast and are thus not likely to be 100% similar. This is probably what causes the <5% dissimilarities you mention.

- P23L767: "0.5 corresponds to climatological forecasts" should be moved to the caption of Fig. 9.

- P24L774-778: it is very difficult to see the blue and red patterns you are mentioning here. Fig. 9d could be improved by, for example, colouring all positive [negative] values with the same shade of blue [red].

- P21L781-784: you already mention this on P23L767-769. The information could be combined.

- The discussion is overall interesting and philosophical in parts. It could however benefit from being shortened in some places, especially Section 4.1, to have more impact.

- P25L807: does "diagram" refer to Fig. 1?

- P25L821-826: these sentences could be removed or shortened as there is no clear point to them. What follows is a clear point.

- P25L842: "might reduce actual skill", as it is not sure.

- P28L950-951: what is the 12 month forecast you are referring to?

- P30L1033-1035: I would suggest removing this sentence as it is a discussion rather than a conclusion and lengthens the conclusion.

- P30L1035-1036: this is especially true at short lead times and could be mentioned here.

- P30L1039-1042: these numbers are specific to lead month 2 and therefore not general enough for the conclusion. It could be removed from the conclusions.

- P30L1043-1045: this information should be moved to the discussion section of this paper.

**Figure 1:**

- Red and green should never be used jointly on a figure (similar comment for Fig. 6 and S3).

- Please specify what the ECMWF hindcasts and the hydrological model are on the figure.

- Change "real observations discharge" to "real discharge observations".

- Add runoff to the model hindcasts and reference run boxes.

- "and Sect. 4.1 for further discussion" can be removed from the caption.

**Figure 5:** "a)" and "b)" are missing from the figure titles.

**Figure 7:** is figure a) really necessary here?

**Supplementary figures:** the text is quite small on Fig. S1.

**Technical corrections:**

- P2L24: "the bias-corrected output".

- P2L25: change "performed" to "produced".

- P2L29: "hindcasting" is not generally used as a verb like it is here. Consider rephrasing to "The skill in the runoff and discharge hindcasts". Same holds in all other instances where this is done in this paper.

- P2L37: "the skill" (2 instances).

- P2L40-41: "Actual skill".

- P2L45: "leads".

- P2L47: "in the following order:".

- P3L73: remove "forecasting model suite he".

- P3L75: "Seasonal hydrological forecasting systems".

- P3L81: change "of" to "from".

- P3L87: "the southern".

- P3L88: "the Alps".

- P3L89: "the Pyreness".

- P4L107: "the NAO".

- P4L108: "Céron".

- P4L110: "the seasonal climate forecast" and "the MAM".

- P4L140: "pseudo-observations".

- P7L233: add ° between the numbers and the letters that refer to the domain extent.

- P7L234: the ")" is missing.

- P7L237: "a three hourly time step" and "The output".

- P7L238: "at a daily resolution".

- P9L312: "except for the Alps".

- P9L322: remove "proper".

- P9L326: "the second for basins".

- P11L403-404: "shortly" does not make sense here, use for example "referred to as".

- P11L414: "river flows".

- P12L454: change "drawn" to "produced" for example.

- P12L457: "3" instead of "2" and "Fig. 2" instead of "Fig. 3".

- P14L489, P15L497, P15L499, P15L501, P15L510: "the skill".

- P15L496, P15L499: "the initialisation".

- P15L511: "soil moisture initialisation".

- P15L512: "the precipitation".

- P16L544: "which lasts until about".

- P17L554: "the left-hand side figure" instead of "At left". Same for "at right".

- P17L558-559: "namely on the left-hand side and the right-hand side".

- P17L571: "implying that there".

- P18L600: "the skill" (2 instances).

- P19L639: remove the extra ".".

- P19L647: "databases".

- P20L655: "b) as a)".

- P20L670: "and for too long".

- P21L671-672: "where VIC reproduced well".

- P21L672: "low flows were".

- P24L783: "the fraction of".

- P26L879: remove extra space before the comma.

- P27L914: "to contradict with our finding that".

- P27L919: add space after "Q33".

- P27L924: "Greuell et al. (2015)".

- P27L926: remove comma after "season".

- P29L973: "Fig. 3".

- P30L1017: "the skill or lack thereof".

- P30L1019: "was determined using the output".

- P30L1025: "mountainous regions".

**References:**

Arnal, L., Cloke, H. L., Stephens, E., Wetterhall, F., Prudhomme, C., Neumann, J., Krzeminski, B., and Pappenberger, F.: Skilful seasonal forecasts of streamflow over Europe?, Hydrol. Earth Syst. Sci. Discuss., https://doi.org/10.5194/hess-2017-610, in review, 2017.

---

## Author Response (AR2)

**Referee 1, Major comments**

1) Naturally, various styles with respect to the use of citations in various parts of a manuscript exist. It is definitely not uncommon to make the introduction a concise review of previous work and to identify therein omissions/open questions that will be addressed in the new paper. Moreover in previous reviews we were actually asked to elaborate more on this. In the introduction we attempted to describe how previous studies generally focussed on smaller (sub-) regions in Europe, on particular seasons (so multiple target months combined) or on selected lead times. So we identify such limitations and from there formulate our own objectives for this paper. In the methods section we describe our system which naturally is a continuation/expansion of previous developments that need a brief description and associated citations. Given the length of the results section we prefer to incorporate descriptive comparisons to findings by others (and thus citations) directly there and leave discussion of possible reasons for consistency or inconsistency with results by others to the discussion section.

2) There is no particular reason to choose April initialisation in Fig.2. May initialisation was chosen in Fig. 6 because in the map some parts of the Danube and the Loire nicely contrast with the surrounding cells, as discussed in the paper.

   We often chose to show and discuss results for lead month 2 and added the following text where Figure 3 is discussed: *because at that lead time approximately 50% of the cells have significant skill.*

   In Figure 5 the number of dashed lines (for specific lead times) was limited to three as more lines make the figures too chaotic. Dashed lines for lead times > 2 months were omitted since these are very close together.

   Regarding the choice of the lead times in Figure 6c we have added in the paper that *the difference in skill between the two variables gradually disappears with increasing lead time*. For lead month 4 the difference is already very small, so it does not make sense to also show results for even longer lead times. More curves would again make the figure chaotic.

3) Regarding Fig. 2 we added: *The same holds for initialisation in other months (see Fig. S1 in the supplementary material), with important exceptions better identified with Fig. 5 and discussed there.*

   Regarding Fig. 3 we added: *Inspection of Fig. S1 leads to the conclusion that to a first approximation regions with skill at other lead times are equal to those listed above for lead month 2 but that the magnitude of skill decreases with increasing lead time as demonstrated in Fig. 2. (keep in mind that a change in lead time corresponds to a change in target time by the same amount). To give an example: for lead month 3 patterns in the skill maps look similar to*

*those provided in Fig. 3 but colours are fainter and target months shift by one month ahead. There are many exceptions to this general rule, e.g. skill due to snow melt that suddenly appears at the end of the melt season at longer lead times while it was not present during the lead months before (see Fig. 5 and the companion paper). A more detailed regional analysis of some of these features is left for future case studies.*

As to Fig. 4, we have replaced the graph for lead month 2 only by a six panel graph showing results for lead months from 1 to 6 and slightly adapted the text describing the conclusions drawn from the figure.

Indeed, Sect. 3.4 compares different metrics for runoff only. We have checked similar figures for discharge and found negligible differences with runoff. We stated this in the text: *We would finally like to note that, while in this sub-section we discussed runoff, we made similar figures and calculations for discharge. Results for these two variables are almost identical.*

4) In our opinion this is a matter of taste and both our strategy and the strategy proposed by the reviewer are good. Each map in Figs. 2 and 3 is summarised in a single point in Fig. 5. The advantage of our approach is that Fig. 5 is easier to understand after first looking at the maps (Fig. 2 and 3). A similar argument holds for Fig. 6.

We have used almost all of the specific comments to improve the paper. For details, see our annotated reply to the word document written by referee #?.

**Referee #2**

- The sentence that explains why skill in discharge exceeds skill in runoff during the first lead month was omitted from the abstract since the abstract was shortened by only mentioning major results. In order to specify what is meant by delay we changed a similar sentence in Sect. 3.2 as follows: *We ascribe this result to the combined effect of the delay between runoff and discharge, with variations in discharge being later in time than the corresponding variations in runoff, and the general tendency of decreasing skill with lead time.*

- The abstract was shortened.

- We have modified the structure of Section 2.1. The first paragraph provides an overview of the system (two types of simulations), the next describes the reference simulation. The topic of paragraphs 3 (overview), 4 (S4) and 5 (bias correction) are the hindcasts. The remaining paragraphs deal with VIC, namely an overview of the simulations (paragraph 6), some settings used in VIC (7) and validation of VIC (8 to 10).

- Fig. 5b. The referee was correct. August should be September. We changed this.

- Fig. 3. Done.

- Sect. 3.3. Reminder was added.

We have used all of the specific comments to improve the paper.

**Editor**

- Better linking the two parts

   In this manuscript results from the "companion paper" are now mentioned 12 times, spread through the manuscript. We particularly like to mention the links that we made in the list of regions with skill in Section 3.1, where we added information on the sources of that skill, a result from the companion paper.

- Comparing performance and forecasting skill
   The first three paragraphs of Section 4.2 deal with this topic and have been rewritten to a large extent:

*There seems to be a broad correspondence between the probabilistic forecast verification presented here and the model validation presented in Greuell et al. (2015) and Roudier et al. (2016). These studies found that average discharge and inter-annual variations therein are well reproduced against observations, consistent with our result that in the first lead month all skill scores, also against real observations (see Fig. S4 for the lead 0 results), are good for large parts of Europe.*

*However, the relation between a model's ability to simulate historic streamflow and its ability to generate skill in seasonal forecasts is complex. There is, for instance, no reason to expect that regions with more theoretical skill than other regions would generally correspond to regions with better historic streamflow simulations. Large model stores of soil moisture and snow tend to lead to more theoretical skill, whether these stores are realistic or not. If they are not realistic, simulations of historic streamflow will be poor, despite the forecast skill. Another example of the problematic relation between validation and verification is that, even in perfect models, regions with small model stores of soil moisture and snow and regions with large interannual variation in precipitation will exhibit small amounts of theoretical and actual skill. So, regions with high quality historic streamflow simulations may for good reasons have little skill in the forecasts.*

*However, what we would expect is that regions of poor model performance have little actual skill (not necessarily little theoretical skill) in the forecasts. In our work, this statement is broadly confirmed by the basins in northern Fennoscandia, which lose much of their skill when using actual instead of pseudo-observations (Fig. 7). In this region VIC indeed performed poorly in reproducing historic flows. Good model performance probably is a necessary (but not sufficient) condition for the generation of actual skill in seasonal forecasts. This is exemplified by some regions with considerable amounts of actual skill in*

*central Europe (e.g. northern part of the Balkans and the Elbe basin in Fig. 7), where VIC's simulations of historic streamflow are much better than in northern Fennoscandia.*